

# A Lagrangian Analysis of the Dynamical and Thermodynamic Drivers of Greenland Melt Events during 1979–2017

Mauro Hermann[1], Lukas Papritz[1], and Heini Wernli[1]

[1]Institute for Atmospheric and Climate Science, ETH Zürich, Zurich, Switzerland

**Correspondence:** Mauro Hermann (mauro.hermann@env.ethz.ch)

**Abstract.**

In this study, we systematically investigate the dynamical and thermodynamic processes that lead to 77 Greenland melt events affecting high-elevated regions of the Greenland Ice Sheet (GrIS) in June–August (JJA) 1979–2017. For that purpose, we compute 8-day kinematic backward trajectories from the lowermost $\sim$500 m above the GrIS during these events. The key

synoptic feature accompanying the melt events is an upper-tropospheric ridge southeast of the GrIS associated with a surface high pressure system. This circulation pattern is favourable to induce rapid poleward transport (up to 40° latitude) of warm ($\sim$15 K warmer than climatological air masses arriving on the GrIS) and moist air masses from the lower troposphere to the western GrIS and subsequently to distribute them in the anticyclonic flow over North and East Greenland. During transport to the GrIS, the melt event air masses cool by $\sim$15 K due to ascent and radiation, which keeps them just above the critical

threshold to induce melting. The thermodynamic analyses reveal that the final warm anomaly of the air masses is primarily owed to anomalous horizontal transport from a climatologically warm region of origin. However, before being transported to the GrIS, i.e., in their region of origin, these air masses were not anomalously warm. Latent heating from condensation of water vapour, occurring as the airstreams are forced to ascend orographically or dynamically, is of secondary importance. These characteristics were particularly pronounced during the most extensive melt event in early July 2012, where, importantly,

the warm anomaly was not preserved from anomalously warm source regions such as North America experiencing a record heat wave. The mechanisms identified here are in contrast to melt events in the low-elevation high Arctic and to midlatitude heat waves, where adiabatic warming by large-scale subsidence is essential. Considering the impact of moisture on the surface energy balance, we find that radiative effects are closely linked to the air mass trajectories and enhance melt over the entire GrIS due to (i) enhanced downward longwave radiation related to poleward moisture transport and a shift in the cloud phase from

ice to liquid primarily west of the ice divide, and (ii) increased shortwave radiation in clear-sky regions east of the ice divide. Given the identified mechanisms that cause extensive melt over the GrIS, the understanding of upper-tropospheric ridges over the North Atlantic, i.e., also Greenland blocking, and its representation in climate models is crucial in determining future GrIS melt and so global sea-level rise.





## 1   Introduction

The Greenland Ice Sheet (GrIS) is the world's second largest ice body holding water equivalent to $6 - 7\,\mathrm{m}$ of global sea level rise (Ridley et al., 2005). Its mass loss due to surface melt and ice discharge has increased strongly over the past 120 years and equaled $286 \pm 20\,\mathrm{Gt\,yr^{-1}}$ in 2010–2018 (Kjeldsen et al., 2015; Mouginot et al., 2019). Not only the current magnitude, but also the speedup of mass loss from the GrIS, observed recently and predicted for the future, are primarily driven by a negative surface mass balance (Enderlin et al., 2014; Van den Broeke et al., 2016). Surface melt has been increasing in the last decades and appears to be the major regulator of the surface mass balance (Box et al.; Fettweis et al., 2012; Andersen et al., 2015; Van den Broeke et al., 2016). At the same time, snow accumulation has decreased since the early 2000s, due to a reduced frequency of cyclones and increased frequency of anticyclones in the close vicinity of Greenland (Chen et al., 2016). Consequently, both contributors to the surface mass balance have favored a stronger mass loss from the GrIS, with melt as the primary factor.

While GrIS melt is highly sensitive to the atmospheric forcing (Hanna et al., 2005), the oceanic forcing during summer melt events is often weak (e.g., Hanna et al., 2014) and the influence of high sea surface temperatures on the GrIS surface mass balance is generally limited due to the katabatic wind blocking effect (Noël et al., 2014). Two largely anti-correlated indices based on geopotential height are often used to capture the dominant modes of variability of the large-scale circulation in the North Atlantic: the North Atlantic Oscillation (NAO) index (Hurrell et al., 2013) and the Greenland Blocking Index (GBI; Hanna et al., 2013). While the NAO captures the strength of the westerly flow over the North Atlantic, the GBI characterises geopotential height anomalies over Greenland with a positive index representative of anticyclonic flow and at times atmospheric blocking. A series of warm summers (June-August, JJA) since the late 1990s were linked to a doubled anticyclone frequency over Greenland compared to the past 50 years (Fettweis et al., 2013). Such anticyclonic conditions are typical for periods with a negative NAO index and positive GBI (NAO-/GBI+). They are characterized by a northward displaced jet stream over Greenland, which favors anomalous meridional transport. This leads to high temperatures in South and West Greenland, but colder conditions and below-average ice loss in Svalbard (Fettweis et al., 2013; Box et al., 2018).

Radiative and turbulent heat fluxes are both known to contribute substantially to melt events over the GrIS (e.g., Fausto et al., 2016; Hofer et al., 2017). Specifically, Hofer et al. (2017) attributed much of the decrease of the surface mass balance during the past 20 years to the GrIS-wide decrease in optically thick clouds, i.e., additional downward shortwave radiation. While clouds block incident shortwave solar radiation (Hofer et al., 2017), they - together with higher water vapour content - tend to enhance downwelling longwave radiation (e.g., Ohmura, 2001; Van Tricht et al., 2016; Gallagher et al., 2018). The net cloud radiative effect caused by these opposed influences depends on cloud properties and the surface albedo and thus varies in sign and magnitude over the GrIS (Box et al., 2012; Wang et al., 2019). Hofer et al. (2019) highlighted the relevance of the cloud liquid water phase in determining the sign of the cloud radiative effect. At Summit, clouds were found to have a warming effect, which was particularly pronounced in summer 2012 (Bennartz et al., 2013; Miller et al., 2015; Van Tricht et al., 2016; Solomon et al., 2017). Furthermore, moist-warm conditions and more importantly liquid clouds are not only instantaneous drivers of





melt, but their effect also accumulates over time to precondition surface melt, on daily (Solomon et al., 2017), seasonal (Park et al., 2015) and annual time scales (Tedesco et al., 2013).


The transport of anomalously warm and humid air masses is a key driver of individual melt events over the GrIS. Warm air implies strong sensible heat fluxes into the ice, such as for example in July 2012 (Hanna et al., 2014; Fausto et al., 2016). During that period, the transport of warm air from a concurrent heat wave over North America (Hoerling et al., 2014) to the GrIS was suggested to be directly related to two melt events (Neff et al., 2014). Additionally, the involved moisture transport

from the western subtropical North Atlantic triggered cloud radiative effects favorable for melt of the GrIS (Neff et al., 2014; Bonne et al., 2015). Optically thin liquid clouds increased the downward longwave radiative flux, still letting shortwave radiation penetrate, and enabled surface melt over the normally dry GrIS inland plateau (Bennartz et al., 2013). In addition to the aforementioned effects, air temperature near the surface directly affects the downwelling longwave radiative fluxes since the bulk of these is emitted in the lowermost kilometer of the atmosphere (Ohmura, 2001). Hence, in this study we will focus on

the mechanisms that lead to the presence of anomalously warm air masses over the GrIS.

Three processes can, in principle, contribute towards the formation of a warm temperature anomaly of airstreams reaching the GrIS (Papritz, 2020), namely the transport of an already warm air mass from a climatologically warmer region towards the GrIS, adiabatic compression during subsidence, and heating by diabatic processes. The latter comprises radiation, latent

heat release in clouds, and turbulent surface fluxes (e.g., Holton and Hakim, 2012). In particular, subsidence is known to be an essential contributor to mid-latitude heat waves (Bieli et al., 2015; Zschenderlein et al., 2019) and warm anomalies in the high Arctic (Ding et al., 2017; Wernli and Papritz, 2018; Papritz, 2020). Furthermore, turbulent surface fluxes over the ocean are typically limited in summer due to the small surface-atmosphere temperature gradient.

Given the high importance of the atmospheric circulation for the variability of Greenland's near-surface temperature, the goal of our study is to improve our understanding of the atmospheric dynamical processes leading to extensive melt episodes over the GrIS. This knowledge is also relevant given the expected increase of GrIS mass loss and global sea level rise, and it might shed light on climate models struggling to simulate the observed circulation anomalies (e.g., Fettweis et al., 2012, 2013). More precisely, this study has two main objectives: First, we want to go beyond case studies and investigate melt events

systematically in the period of 1979–2017. Still, the well-studied and most extreme melt event of July 2012 will serve as an excellent example to illustrate our methods and findings. Second, we aim to investigate the history and thermodynamic evolution of air masses associated with Greenland melt events with the aid of Lagrangian backward trajectories. This approach will enable us to answer the following questions:

Q1) How often did melt events occur over the GrIS during 1979–2017?

Q2) What is the synoptic flow configuration and the air mass pathways during melt events?

Q3) Which thermodynamic air mass modifications and radiative effects over the GrIS caused these melt events?





Q4) Does the answer to Q2 and Q3 differ for subregions of the GrIS?

## 2 Data and Methods

### 2.1 ERA-Interim data

This study is based on ERA-Interim reanalysis data from the European Centre for Medium-Range Weather Forecasts (ECMWF; Dee et al., 2011). The data is available every 6 h from 1979 to 2017, on 60 vertical levels and interpolated to a grid with a horizontal grid spacing of $1°$. The reanalysis data serves as best estimate of the past atmospheric state on the synoptic scale, which is why we implicitly refer to it as the actual state of the atmosphere. As climatologies of the variables used for Eulerian analyses (Table 1), we compute 10 d-averages of the 6-hourly data centered on the respective calendar day over the entire

period 1979–2017 (i.e., the long-term average of 39 x 41 time steps). Note that for radiation, we use fields with the same time of day only to account for the daily cycle (i.e., 39 x 11 time steps). We use the ice outline after Zwally et al. (2012) to separate ice from land grid cells in Greenland. Only grid cells with a center inside the ice outline are classified as ice grid cells, which leads to 519 ice grid points in ERA-Interim, corresponding to a GrIS area of 1.73 million $km^2$, which is slightly larger (+0.7%) than observed (Zwally et al., 2012).

### 2.2 Melt event definition

As previous studies focused on single Greenland melt events, such as in July 2012 (e.g., Nghiem et al., 2012; Bennartz et al., 2013; Tedesco et al., 2013; Neff et al., 2014; Bonne et al., 2015), there is yet no generally accepted definition of a melt event for climatological studies. We define them as follows: The occurrence of surface melt is approximated by a skin temperature ($SKT$) greater or equal to $-1°C$, as in earlier studies (e.g., Nghiem et al., 2012). A time step is interpreted as part of a melt

event if at least 5% of the total GrIS surface area is melting and including grid points above 2000 m elevation ("melt time

**Table 1.** The ERA-Interim variables used for the evaluation of the synoptic situation over the GrIS ("Eulerian variables").

| **Eulerian variables** | | |
|---|---|---|
| **abbreviation** | **variable name** | **unit** |
| *RR* | 6 h-accumulated rainfall | [mm] |
| *SF* | 6 h-accumulated snowfall | [mm] |
| *SSR(D)* | (downward) surface shortwave radiation | [W m$^{-2}$] |
| *STR(D)* | (downward) surface thermal radiation | [W m$^{-2}$] |
| *TCVHT* | total column horizontal water vapour transport | [kg m$^{-1}$ s$^{-1}$] |
| *TCV* | total column water vapour | [kg m$^{-2}$] |
| *TCIW/TCLW* | total column ice/liquid water | [kg m$^{-2}$] |
| *Z500* | 500 hPa geopotential height | [m] |
| $\theta_{10m}$ | potential temperature on the lowest model layer ($\sim$10 m) | [K] |





**Table 2.** Average (Avg.), standard deviation ($\sigma$), minimum (Min.) and maximum (Max.) of melt event duration, maximum elevation experiencing surface melt (ME) and maximum two-meter temperature (*T2M*) at this highest grid point, respectively, as well as minimum and maximum melt extent during the melt event. For *T2M@ME*, the elevation at which it was observed is indicated in brackets. The total number of melt events is $N = 77$.

| | Dur. [d] | ME [m] | Max. T2M @ME [°C] | Min. melt extent [%] | Max. melt extent [%] |
|---|---|---|---|---|---|
| **Avg. $\pm\,\sigma$** | $4.1 \pm 3.4$ | $2692 \pm 193$ | $-0.2 \pm 1.3$ | $8.7 \pm 5.2$ | $44.6 \pm 10.7$ |
| **Min.** | 1.25 | 2333 | -2.6 [2826 m] | 1.2 | 29.0 |
| **Max.** | 16.25 | 3175 | +5.0 [2637 m] | 25.6 | 94.8 |

step"), to distinguish melt events from the typical summer melt in the GrIS ablation area. In order to avoid splitting of a - from a dynamical point of view - coherent melt event due to the pronounced diurnal temperature cycle, we include non-melt time steps when identifying coherent melt events. This is done as follows: intermediate non-melt and melt time steps are connected in time to yield melt events with the starting (end) date defined as the first (last) time step when melt was detected, but not

preceded (followed) by melt for more than 24 hours. The thresholds of 5% and 2000 m were chosen with hindsight, such that a reasonable maximum melt event duration of around two weeks and a sufficiently large sample size of 77 melt events results. Events shorter than 24 hours are neglected.

The 77 Greenland melt events in 1979–2017 (Tables S1–S3) lasted between 1.25 d and 16.25 d and on average $4.1 \pm 3.4$ d

(Table 2). Surface melt during short events typically covered around a third of the GrIS at maximum. On average, about half ($44.6 \pm 10.7\%$) of the GrIS was melting at the time of maximum extension of the event. The three melt events affecting the largest ice area were EV69 (94.8%), EV35 (83.9%) and EV70 (70.3%) in early July 2012, June 2002 and end of July 2012, respectively. EV69 is the most closely investigated melt event in the literature, where surface melt occurred up to the highest ERA-Interim grid point at 3175 m. Considering all events, the maximum elevation with surface melt was $2692 \pm 193$ m. The

maximum two-meter temperature at the most elevated grid point experiencing melt averaged slightly below 0°C.

## 2.3 Trajectories

In addition to the classical way of depicting atmospheric phenomena from the Eulerian perspective, we use the Lagrangian framework to investigate air mass modifications, the underlying physical processes and general flow structures. The Lagrangian analysis tool LAGRANTO (Wernli and Davies, 1997; Sprenger and Wernli, 2015) basically solves the trajectory equation

(Eq. 1) numerically.

$$\frac{D\mathbf{x}}{Dt} = \mathbf{u}(\mathbf{x}) \tag{1}$$

where $\mathbf{x}$ is the position of an individual air parcel and $\mathbf{u}$ the 3D wind vector. We use 3D ERA-Interim wind fields to calculate kinematic backward trajectories from pre-defined starting locations and trace a set of variables along the trajectories (Table 3).





**Table 3.** The ERA-Interim variables traced along the 8-day backward trajectories ("Lagrangian variables").

| Lagrangian variables | | |
| --- | --- | --- |
| **abbreviation** | **variable name** | **unit** |
| $MASK$ | ice mask of the GrIS [0,1] | [ ] |
| $T$ | temperature | [°C] |
| $\theta$ | potential temperature | [K] |
| $\theta_{cl}$ | 1979–2017 potential temperature climatology | [K] |
| $SKT$ | skin temperature | [K] |
| $q$ | specific humidity | [g kg$^{-1}$] |

In the domain defined by the 519 ice grid points (Sect. 2.2) trajectories are started equidistantly every 80 km in the horizontal.
In the vertical, trajectories start at 20, 40 and 60 hPa above ground level, i.e., from the lowermost ∼500 m of the atmosphere, resulting in $3 \times 267$ starting points. The trajectories are calculated eight days backward in time and start every 6 h during a melt event. For smoother plotting, trajectory positions and all variables are written out every 3 h along the trajectories. One application of the trajectories is to perform so-called Lagrangian forward projections (LFP; Liniger and Davies, 2003; Sodemann et al., 2008), i.e., certain properties of the air mass, such as for example the total 8-day adiabatic warming, are
projected onto the trajectory starting point above Greenland.

## 2.4   Lagrangian evaluation of thermodynamic energy equation

We evaluate the thermodynamic energy equation in order to get insight into the warming mechanisms along trajectories.

$$\frac{DT}{Dt} = \frac{\kappa T \omega}{p} + H \left( \frac{p_0}{p} \right)^{-\kappa} \tag{2}$$

According to Holton and Hakim (2012) and Bieli et al. (2015), the relationship between temperature, vertical motion, and
diabatic processes (Eq. 2) follows from the thermodynamic energy equation and the material derivative of potential temperature $\theta = T(p_0/p)^\kappa$ [K], with reference pressure $p_0$ and $\kappa = R/c_p = 0.286$. The total diabatic heating rate is $H = D\theta/Dt$ [K s$^{-1}$], and the vertical velocity equals $\omega = Dp/Dt$ [Pa s$^{-1}$].

$$\Delta T = T_{adi} + T_{diab} \tag{3}$$

We split the warming integrated along the eight-day trajectories, $\Delta T$, into adiabatic (Eq. 2, 1st term on r.h.s.) and diabatic
(Eq. 2, 2nd term on r.h.s.) components (Eq. 3). The diabatic change of temperature along the trajectory is calculated from $\theta$ and $p$ with the numerical approximation in Eq. 4. The adiabatic warming then follows as a residuum from this term and the total $\Delta T$ along the trajectory (Eq. 3).

$$\Delta T_{diab} = \sum_{t \in \{-189\,\mathrm{h}, ..., -3\,\mathrm{h}, 0\,\mathrm{h}\}} \frac{\theta(t) - \theta(t - 3\,\mathrm{h})}{3\,\mathrm{h}} \left( \frac{p_0}{0.5 \cdot (p(t) + p(t - 3\,\mathrm{h}))} \right)^{-\kappa} \tag{4}$$





Adiabatic warming/cooling is a consequence of adiabatic compression/expansion due to vertical motion ($\omega$). We expect
diabatic heating ($H$) to be dominated by radiative clear-sky cooling at $\sim 1\,\mathrm{K\,d^{-1}}$ (Cavallo and Hakim, 2013; Papritz and
Spengler, 2017) and latent heating/cooling by condensation of water vapor or evaporation/sublimation of hydrometeors in and
below clouds. Oceanic surface sensible heat fluxes in the midlatitudes are typically reduced in summer compared to winter and
can only affect the few airstreams travelling in the surface layer.

## 3    Detailed analysis of melt event EV69

In order to illustrate our methodology and the processes at play, we start with a detailed case study of the EV69 melt event. It
lasted from 18 UTC 2 July to 18 UTC 17 July 2012 and included the most extreme period of surface melt in terms of elevation
(up to the highest grid cell at 3175 m) and maximum coverage (94.8%) on 12 July.

### 3.1    Synoptic situation

The synoptic flow configuration during EV69 was characterized by an exceptionally strong 500 hPa geopotential height
anomaly, $Z500'$, over and near Greenland (Figs. 1a, c), and can be divided in two distinct periods of about one week each.
The first period was initiated by the deepening of a slowly propagating low pressure system near Newfoundland (not shown)
and an amplifying upper-level ridge over the Central North Atlantic. During the subsequent eight days, from 18 UTC 2 July
to 12 UTC 10 July, the North Atlantic circulation pattern resembled a typical Omega-blocking (e.g. Woollings et al., 2018), as
evident from the shape of the $Z500$ contours (Fig. 1a). High values of $Z500$ located southeast of Greenland, with an anomaly
vastly above the 90th percentile, were sustained and stabilized by an upstream trough over Newfoundland and a downstream
trough over the UK. Another strong ridge was present further downstream over Russia (Fig. 1a). In the lower troposphere, the
southerly flow between the upstream low and the Greenland ridge advected exceptionally warm air to southern and western
Greenland, causing near-surface potential temperature anomalies, $\theta'_{10\mathrm{m}}$, of more than 5 K, as well as to neighbouring regions
such as Newfoundland, the Labrador Sea and Baffin Bay (Fig. 1b). Another striking feature is the exceptional heat wave of
similar anomaly magnitude over the Great Plains of North America (Hoerling et al., 2014; Neff et al., 2014).

By the end of the first period, the positive $Z500'$ migrated poleward. Thus, the circulation during the second half of EV69,
the 7.25-day period from 18 UTC 10 July to 18 UTC 17 July, was characterized by a less meridional flow south of Greenland
and a cut-off anticyclone centered over the GrIS, while a deep trough dominated over northern Europe (Fig. 1c). The median
$Z500'$ over Greenland during this period was around +150 m (>90th percentile), except for the southern tip of the GrIS. It went
along with an equally exceptional $\theta'_{10\mathrm{m}}$ over the GrIS and the northeastern North Atlantic, peaking in northern Greenland with
values of >+7 K (Fig. 1d).

The combination of the cyclone over Newfoundland and the anticyclone over the GrIS favoured northward transport of
low-level air masses from the subtropical North Atlantic towards the southern tip of Greenland and the Labrador Sea. Figure



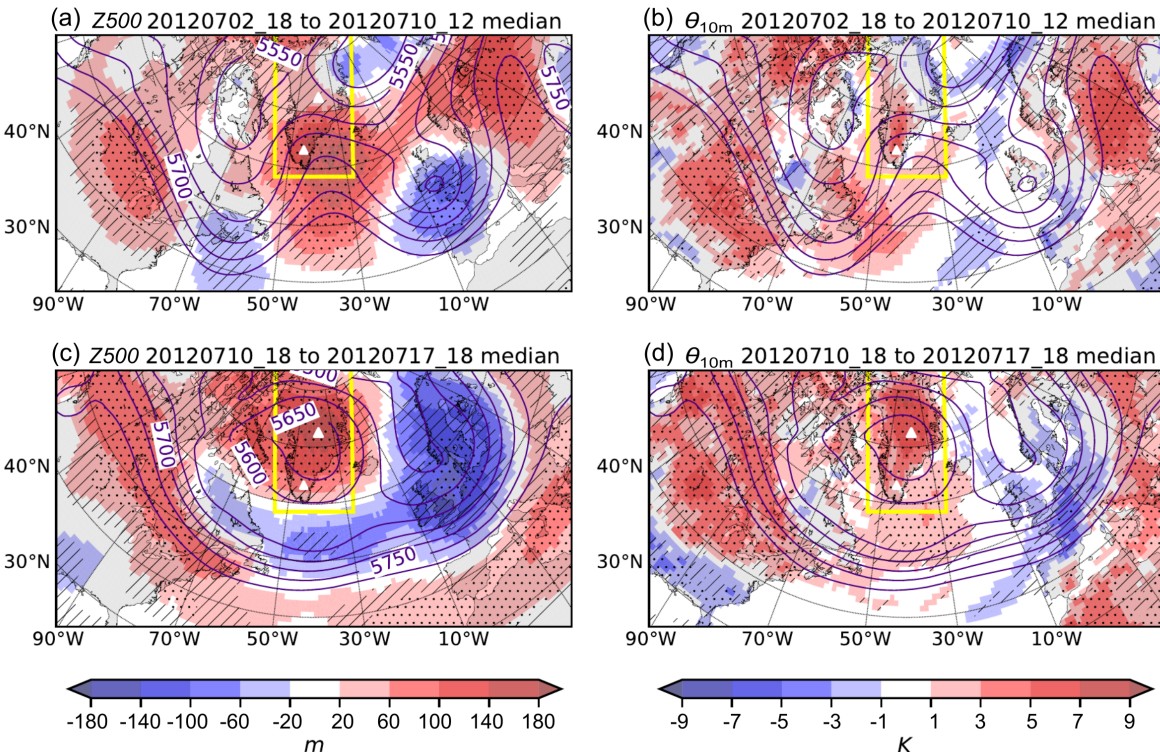

**Figure 1.** The median fields of 500 hPa geopotential height (*Z500*; contours) for the two periods during EV69 (a, b) 18 UTC 2 July to 12 UTC 10 July (8 days) and (c, d) 18 UTC 10 July to 18 UTC 17 July (7.25 days). Median anomalies of the synoptic fields in the respective periods are shown in colors (a, c) *Z500* and (b, d) near-surface potential temperature ($\theta_{10m}$). Hatching (stippling) indicates an anomaly outside the 25–75 (10–90) percentile range of equally long periods in JJA 1979–2017. The yellow box indicates the location of the GrIS.

2 shows air masses arriving over the GrIS during melt time steps - hereafter referred to as "melt air masses". Several distinct streams of melt air masses can be identified: Two airstreams originating close to the east coast of North America ascended over the southern tip of Greenland into the mid-troposphere and descended anticyclonically onto the central and eastern GrIS (labels C, E). Another important contribution stemmed from the subtropical North Atlantic. These airstreams followed a northward trajectory also towards southern Greenland, where the higher air masses ascended slightly to reach Southdome in a straight trajectory (S), whereas those in the marine boundary layer moved into the Labrador Sea and from there - remaining at low altitudes - further northward to reach the northern GrIS in a final rapid ascent (N1). It is interesting to note that the bulk of N1 air parcels did not ascend along Greenland's west coast but instead remained at low levels until they approached Northwest Greenland. An additional airstream reached the North at lower altitudes from neighbouring regions and approached the GrIS from the Northwest (N2) or descended in the dry intrusion of the low near Newfoundland. Hereafter, we refer by C, E, N, S, or W air masses to air masses arriving in the specific region, irrespective of whether they follow a similar trajectory as shown in the previsouly discussed example (Fig. 2).



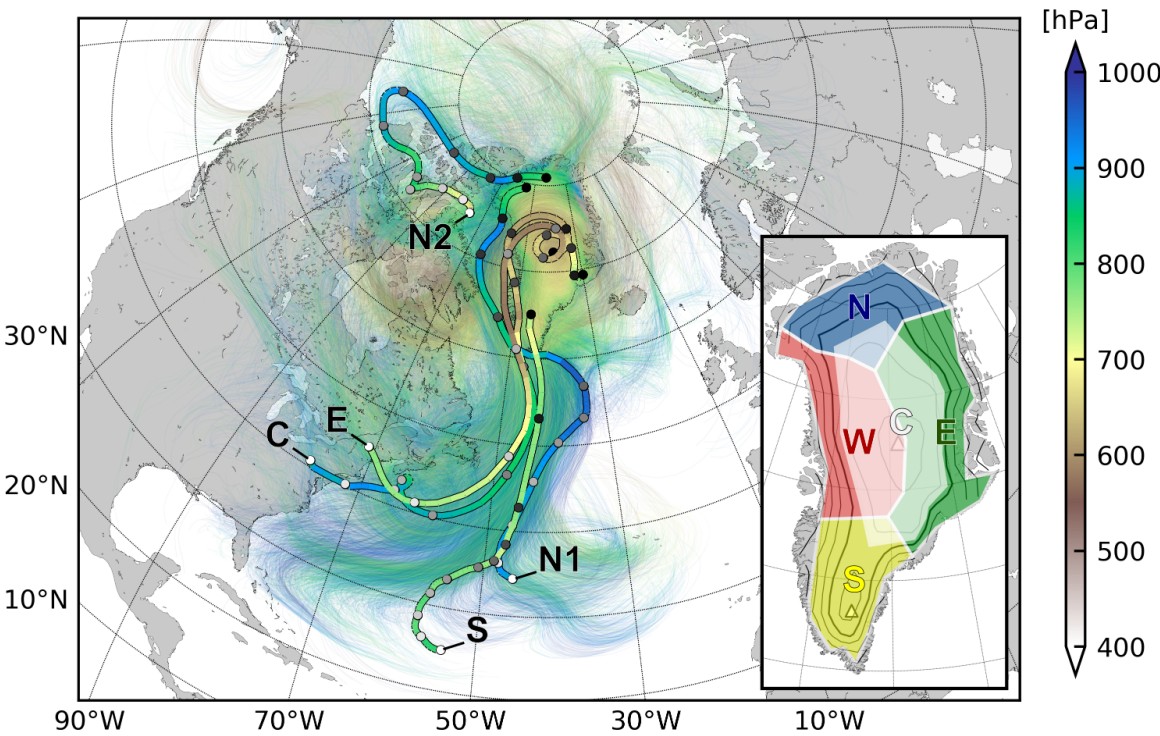

**Figure 2.** All extended (10-day) melt trajectories of EV69 colored according to their pressure. Five representative example trajectories represent characteristic airstreams (S, C, E, N1, N2), shown in thicker lines with one circle per day colored from white ($t = -240\,\text{h}$) to black ($t = 0\,\text{h}$). The subfigure map shows characteristic regions of the GrIS: South (S), West (W), North (N), East (E), and central plateau (C).

## 3.2 Lagrangian forward projection

The Lagrangian analysis is split in two parts in which we identify (i) sources of air masses (Sect. 3.2.1) and (ii) mechanisms
(Sect. 3.2.2) that contributed to surface ice melt over the GrIS prior to or during EV69; specifically, we consider the characteristics of the air mass origin and the following transport focusing on thermodynamic temperature changes along the trajectories.

### 3.2.1 Air mass origin

By comparing with the climatological characteristics of air masses arriving over the GrIS, we pinpoint the anomalous nature of the EV69 melt trajectories in terms of latitude, altitude and temperature anomaly. For each melt trajectory we define the relative
minimum latitude and relative minimum pressure as the differences between the respective values of latitude and pressure at the destination over the GrIS and the minimum values along the trajectory, thus, indicating the largest changes in latitude and pressure. To that end, we make use of 8-day backward trajectories from the lowermost 60 hPa above the GrIS (see Sect. 2.3) during melt time steps associated with EV69. Figures 3a, c show the relative minimum latitude and pressure projected onto the



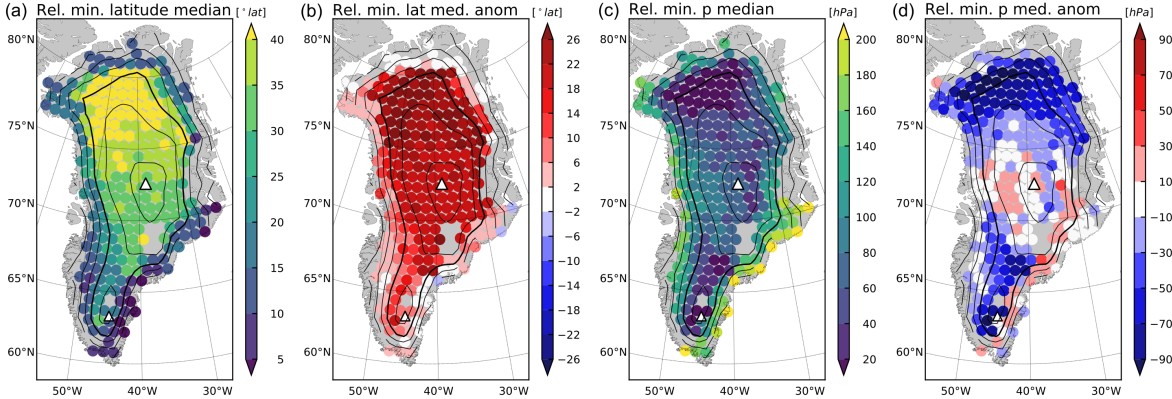

**Figure 3.** Lagrangian forward projection (LFP) of the median (a) relative minimum latitude, (c) relative minimum pressure (where positive values mean from (a) lower latitudes and (c) above), and (b, d) the respective anomaly fields of 8-day melt trajectories of EV69 wrt. the climatological summertime air parcels. The contours indicate elevation in 500 m intervals with the 2000 m isoline in solid. Summit and Southdome are marked with triangles.

trajectory starting locations over the GrIS in a so-called Lagrangian forward projection. In addition, we show their anomalies

with respect to the climatological reference defined as all air parcels that arrived over the GrIS during JJA 1979–2017, i.e., typical values for all summertime air masses that arrive over the GrIS (Figs. 3b, d).

Generally speaking, the median melt air mass moved poleward by about 20° latitude in region S and up to 40° latitude in region N (Fig. 3a). Thus, it originated from a region in the atmosphere located around 20° latitude further south wrt.

climatology (Fig. 3b). In addition, it descended by about 50 hPa less than the climatological air parcels, which applies, in particular, to Southdome and the region N (Fig. 3d). In fact, many melt air masses arriving in these regions show a relative minimum pressure of less than 20 hPa, meaning that they had never reached far above their final elevation and hence ascended from the lowermost parts of the troposphere (Fig. 3c). Further, most air parcels show initially, at $t = 192$ h, a small potential temperature anomaly wrt. the local climatology, $\theta_{cl}$, of ~0.5–2.5 K, which is highly unusual compared to the average air mass,

which shows an approximately 0 K initial anomaly (Fig. S1c). The origin of these initial warm anomalies is related to the North American heat wave (Hoerling et al., 2014) and other anomalously warm source regions, predominantly in the Canadian Arctic. An interesting exception concerns melt air masses reaching region N, which did not come from regions with a positive temperature anomaly. They, however, show the largest relative minimum latitude, i.e., strongest meridional transport, of more than 40° latitude (Fig. 3a). Other exceptional melt air masses arrived over regions C and E, showing slightly increased descent

compared to the climatological air masses (Fig. 3d).



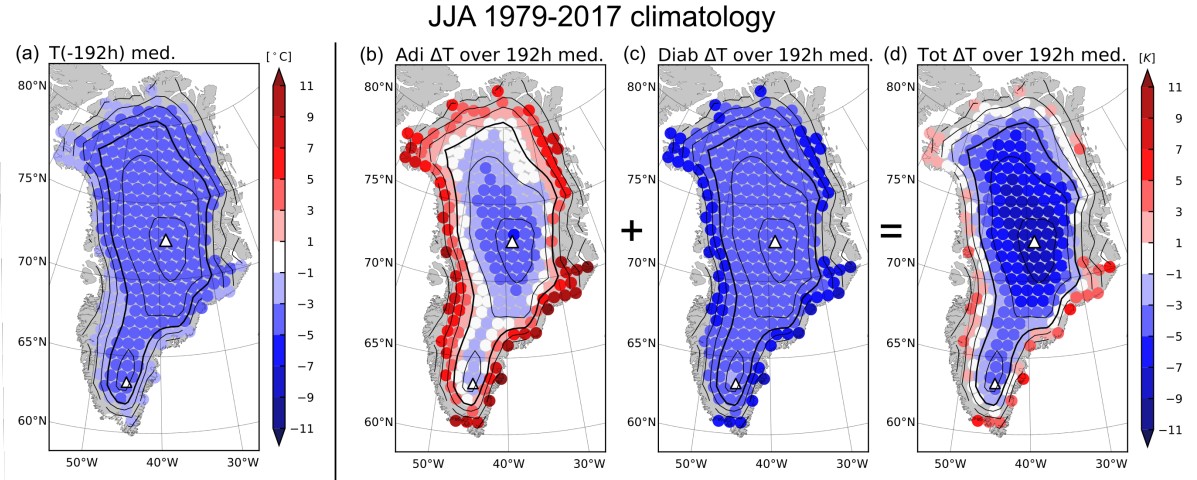

**Figure 4.** LFP maps as in Fig. 3 of the (a) initial temperature at $t = -192\,$h, and (b) adiabatic, (c) diabatic, and (d) total temperature change over eight days and of the JJA 1979–2017 climatological summertime air masses.

### 3.2.2 Air mass evolution

In order to assess the relative importance of adiabatic and diabatic temperature changes for the final temperature anomalies of EV69 melt air masses, we first consider the typical temperature evolution of all air parcels that arrive over the GrIS during JJA 1979–2017. The initial temperature is very uniform for all trajectories with a median of $T = -3.8°$C at $t = -192\,$h
(Fig. 4a). While in the following all air parcels experience diabatic cooling of about 0.5–1 K day$^{-1}$ (Fig. 4c), mostly due to longwave radiative fluxes, the adiabatic temperature changes exhibit a strong elevation dependence. Specifically, C air masses cool adiabatically, indicating ascent prior to arrival over the GrIS (Fig. 4b). Air masses arriving in the coastal areas, in contrast, experience overall adiabatic warming. This descent is likely the result of katabatic drainage flows prevalent over the GrIS (cf. Heinemann and Klein, 2002). Consequently, cooling dominates the temperature evolution of most air masses arriving over the
elevated regions of the GrIS, whereas at lower elevations adiabatic warming compensates for much of the radiative cooling such that these air masses experience little to slightly positive overall temperature changes (Fig. 4d). Except for some of the latter, summertime air parcels arrive with negative temperatures within the lowermost 60 hPa ($\sim$500 m) aloft the GrIS, because of $T < 0°$C at $t = 192\,$h (Fig. 4a) and cooling during the transport to Greenland (Fig. 4d).

The adiabatic and diabatic temperature modifications of EV69 melt air masses deviated from the typical summer air mass as shown in Fig. 4. At the start of their trajectory ($t = -192\,$h), melt air masses were about 10–16 K warmer than climatological air masses (Fig. 5a). Melt air masses arriving closer to the coast of Greenland showed a smaller initial temperature anomaly ($T'<+10\,$K) than those arriving in region C ($T'>+16\,$K). From the evolution during the subsequent eight days, we again refer to representative melt airstreams with different characteristics (see Fig. 2): a melt airstream arriving over (i) the central plateau



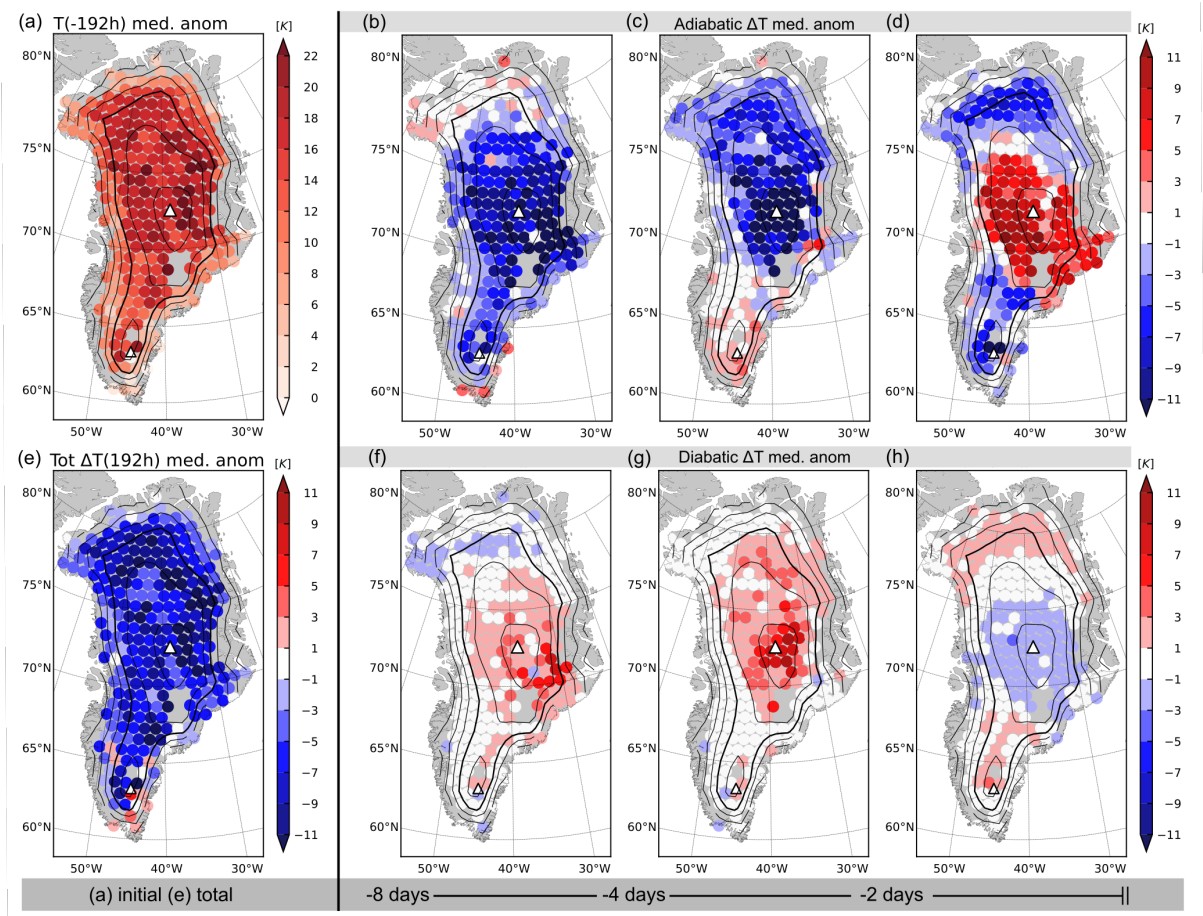

**Figure 5.** LFP anomaly maps of the (a) initial temperature at $t = -192\,\mathrm{h}$, and (b–d) adiabatic, (f–h) diabatic, and (e) total temperature change over eight days wrt. the climatological summertime air masses shown in Fig. 4. The adiabatic and diabatic temperature change anomalies are split up in the periods (b, f) $t = -192\,\mathrm{h}$ to $t = -96\,\mathrm{h}$, (c, g) $t = -96\,\mathrm{h}$ to $t = -48\,\mathrm{h}$, and (d, h) $t = -48\,\mathrm{h}$ to $t = 0\,\mathrm{h}$.

and East Greenland ("C" and "E"), (ii) North Greenland ("N1", "N2"), and (iii) South Greenland (including Southdome; "S"), respectively. Figures 5b–d and f–h show the adiabatic and diabatic temperature changes during days 0–4, 5–6, and the last two days of the transport to Greenland. N air masses exhibited a thermodynamic evolution in the beginning that was close to that of the climatological air masses (Figs. 5b, f) and an orographically induced final ascent reflected in enhanced adiabatic cooling (Figs. 5c, d). In close proximity to the GrIS, N air parcels ascended from the South (N1 in Fig. 2) or the West (N2 in Fig. 2) to

the northern GrIS. Airstream S showed a similar pathway as N1 but with initially exceptional adiabatic cooling before reaching Greenland (Fig. 5b) and subsequently orographic ascent from the South (Fig. 5d). The associated adiabatic cooling and diabatic warming were less pronounced and occurred later in S compared to N (Figs. 5c, d, g, h).



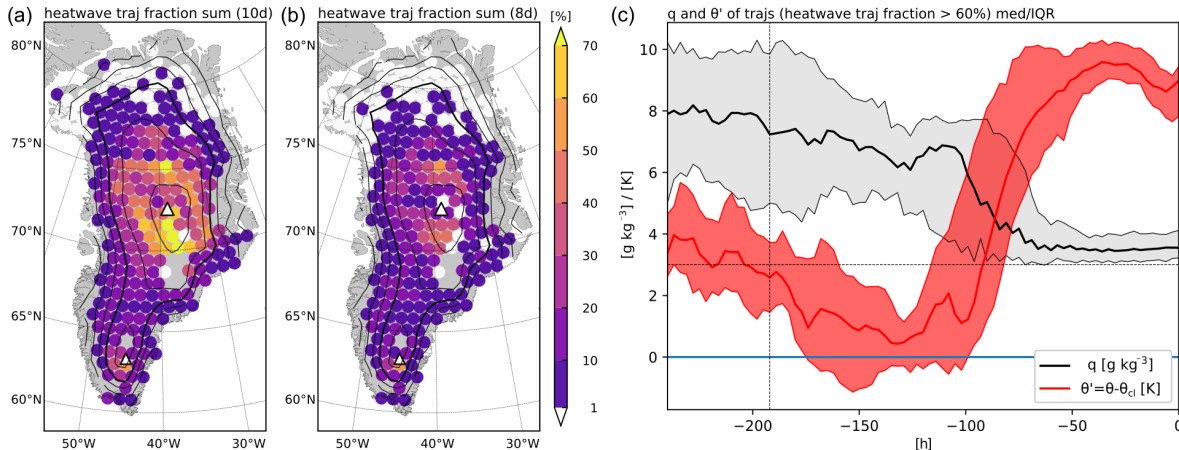

**Figure 6.** LFP maps of the trajectory fraction being associated with the North American heat wave (see text for details), for (a) the extended 10-day or (b) 8-day trajectories; (c) shows the median (solid line) and inter-quartile range (shading) of specific humidity $q$ (grey) and $\theta'$ wrt. $\theta_{cl}$ (red) of heat wave trajectories ending at locations with a heat wave trajectory fraction $>60\%$. Dashed lines in (c) indicate $t = -192\,\mathrm{h}$ and $\theta' = +3\,\mathrm{K}$.

Airstreams C and E experienced much stronger than usual adiabatic cooling during most of the 8-day period, especially be-
tween $t = -144\,\mathrm{h}$ and $-48\,\mathrm{h}$, which indicates enhanced ascent (Figs. 5b, c). This ascent either stemmed from dynamical lifting at the polar front, i.e., by the trough over Newfoundland (C in Fig. 2), and/or orographic lifting at the southern tip of Greenland (E in Fig. 2; cf. Stohl, 2006). Interestingly, during the final 48 h, these airstreams experienced strong adiabatic warming due to descent onto the GrIS. Over eight days, the total adiabatic warming anomaly is typically below $-10\,\mathrm{K}$ and diabatic heating in melt air masses is anomalous by $+5\,\mathrm{K}$ - both signals especially distinct in airstreams C and E. The total temperature change was uniform over the GrIS and its GrIS-wide median equaled $-6.2\,\mathrm{K}$ (Fig. 5e), with the initially warmest air masses (Fig. 5a) cooling more than the rest. Nevertheless, 40–80% of the initial warm temperature anomaly of the melt air masses wrt. the clima-
tological air masses (Fig. 5a) was sustained and not compensated by stronger cooling than along the climatological trajectories.

Previous studies (Neff et al., 2014; Bonne et al., 2015) have shown that anomalously warm and humid air masses associated with the heat wave over North America contributed to EV69. To quantify this contribution, we extend our set of backward trajectories to ten days and define heat wave trajectories as those melt trajectories that (i) pass the North American Great Plains at some point, and (ii) at the same time have a potential temperature anomaly wrt. $\theta_{cl}$ in excess of 3 K. The fraction of these trajectories among all melt trajectories is shown in Figs. 6a, b. The highest contributions of heat wave air masses are found for airstreams C and E (up to 70% near Summit; C and E in Fig. 2). The heat wave trajectories ending at locations with more than 60% U.S. heat wave contribution were initially anomalously warm (also constrained by the selection criteria) and had a high specific humidity of $\sim 8\,\mathrm{g\,kg^{-1}}$ (Fig. 6c). During their way from the American continent to the western North Atlantic (until





$t = -120\,\mathrm{h}$), however, these air masses lost most of their warm anomaly. This is in contrast to the idea that they carried the original temperature anomaly from North America to Greenland. At the same time, they conserved their moisture to a large degree, with the driest ones picking up additional moisture once they reached the open ocean. It is only between $t = -120\,\mathrm{h}$

to $-48\,\mathrm{h}$ when the air masses rapidly moved poleward into a climatologically much colder region that their warm anomaly increased from around +1 K to almost +10 K. The concurrent reduction of specific humidity confirms the condensation of water vapour and aforementioned diabatic warming during that period (Figs. 5f, g), which was linked to ascent along the sloping isentropes as the air mass moved poleward. Indeed, air masses came from the U.S. heat wave but the temperature anomaly near the origin of their trajectories was not directly responsible for the warm anomaly upon arrival in Greenland.


In summary, we conclude that exceptional poleward transport and ascent of relatively moist and climatologically warm air masses contributed substantially to EV69 by advection towards the climatologically much colder GrIS region. Nevertheless, air masses leading to this Greenland melt event were cooled stronger than usual during their transport to the GrIS. In these warm, moist and poleward ascending airstreams, cloud formation and latent heating compensated some of the adiabtic cooling

and contributed to the warm anomaly over the GrIS.

### 3.3   Linkage to clouds and radiative effects

The characteristic airstreams during EV69 (C, E, N1, N2, S; Fig. 2) likely played an important role in modulating the spatial distributions of rain and snowfall, cloud liquid water, and radiative fluxes over the GrIS, which in turn had a strong impact on the melt potential. To illustrate these inter-linkages, we consider rain rate (*RR*), total column liquid water (*TCLW*), surface short-

wave downward radiation (*SSRD*) and the sum of net surface short- and longwave (thermal) radiation (*SSR+STR*; Figs. 7a–d), as well as their anomalies wrt. the 1979–2017 summer climatology (Figs. 7e–h) for melt time steps at the respective grid points.

Figure 7 indicates distinctive patterns over the GrIS that are related to the pathways of and processes within air masses arriving on the GrIS. The ascending airstreams S, N1, and N2 were associated with strong anomalies of *TCLW* and rainfall in

southern and north-western Greenland, respectively (Figs. 7a, b, e, f). In particular, airstream N1 carried moisture far northward and, as it ascended onto the GrIS north of Summit, the air parcels reached saturation. Consequently, *TCLW* and *RR* exceeded their climatologies in the entire region N. Similarly, also region S experienced strong *TCLW* and *RR* anomalies due to the ascending airstream S. Interestingly, in region W, *TCLW* and *RR* were below average. This is related to the fact that the humid air masses associated with airstream N1 remained in the boundary layer and did not ascend until they reached northwestern

Greenland. Consequently, the air masses arriving in region W were transported at higher levels than airstream N1 and did not ascend or may even have experienced slight descent, thus leading to cloud-free conditions. Regions with precipitation experienced an additional heat flux into the ice from rain, while snowfall and total column ice water were widely reduced (Fig. S2b), as was previously highlighted by Doyle et al. (2015) and Fausto et al. (2016). In direct relation to increased *TCLW*, *SSRD* was strongly reduced, especially in region N (Figs. 7c, g). In contrast in regions E and W, clear-sky conditions with hardly any pre-

cipitation prevailed (Fig. 7a), as evident from extremely low *TCLW* (Fig. 7b), as well as substantially increased *SSRD* (Fig. 7g).



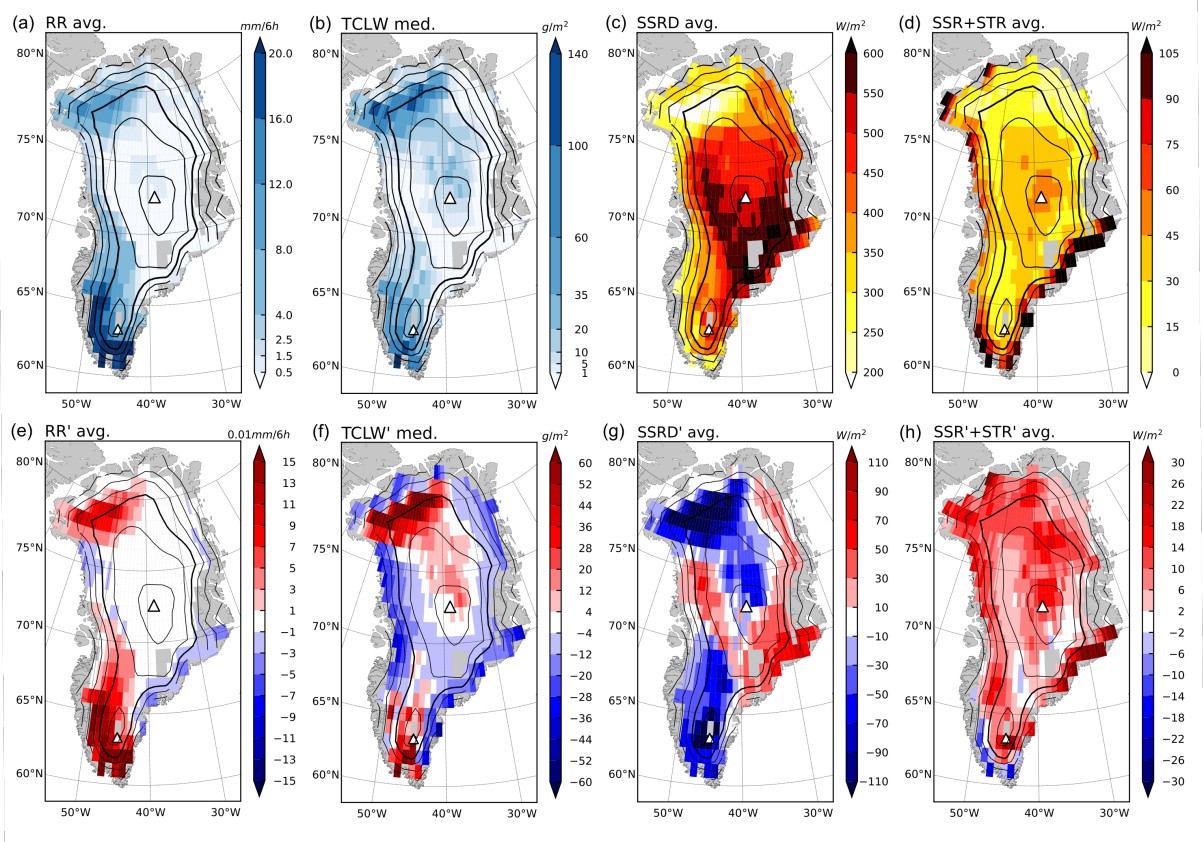

**Figure 7.** Composites of (a) rainfall (*RR*), (b) total column liquid water (*TCLW*), (c) surface shortwave downward radiation (*SSRD*), (d) net surface long- and shortwave radiation (*SSR+STR*) during EV69 melt time steps, and (e–h) the respective anomalies wrt. climatology. Depending on the melt impact of the variable, the composite results from the average (*RR, SSRD, SSR+STR*) or the median (*TCLW*). The contours indicate elevation in 500 m intervals with the 2000 m isoline in solid. Summit and Southdome are marked with triangles.

Southeast of the plateau, this relates to the branch of the descending airstreams C and E that stretched anticyclonically from west to east across the central portion of the GrIS.

Despite the reduction in downwelling shortwave radiation in the regions with above average *TCLW*, the sum of net short-
and longwave radiation was positive across the entire GrIS (Fig. 7d) and almost everywhere in excess of climatological values (Fig. 7h). This is in part explained by enhanced downwelling longwave radiation in the cloudy regions. There and also at and north of Summit, the shift of the cloud phase to the liquid regime was found to be decisive for the observed melt (Bennartz et al., 2013; Solomon et al., 2017). In addition, also total column water vapour was above the long-term summer climatology over all of the GrIS (area-weighted average anomaly of +3.3 kg m$^{-2}$; Fig. S2f). The area-weighted GrIS-wide
average anomaly of *SSR+STR* (Fig. 7h) amounted to +9.4 W m$^{-2}$ and provided additional energy corresponding to a melting



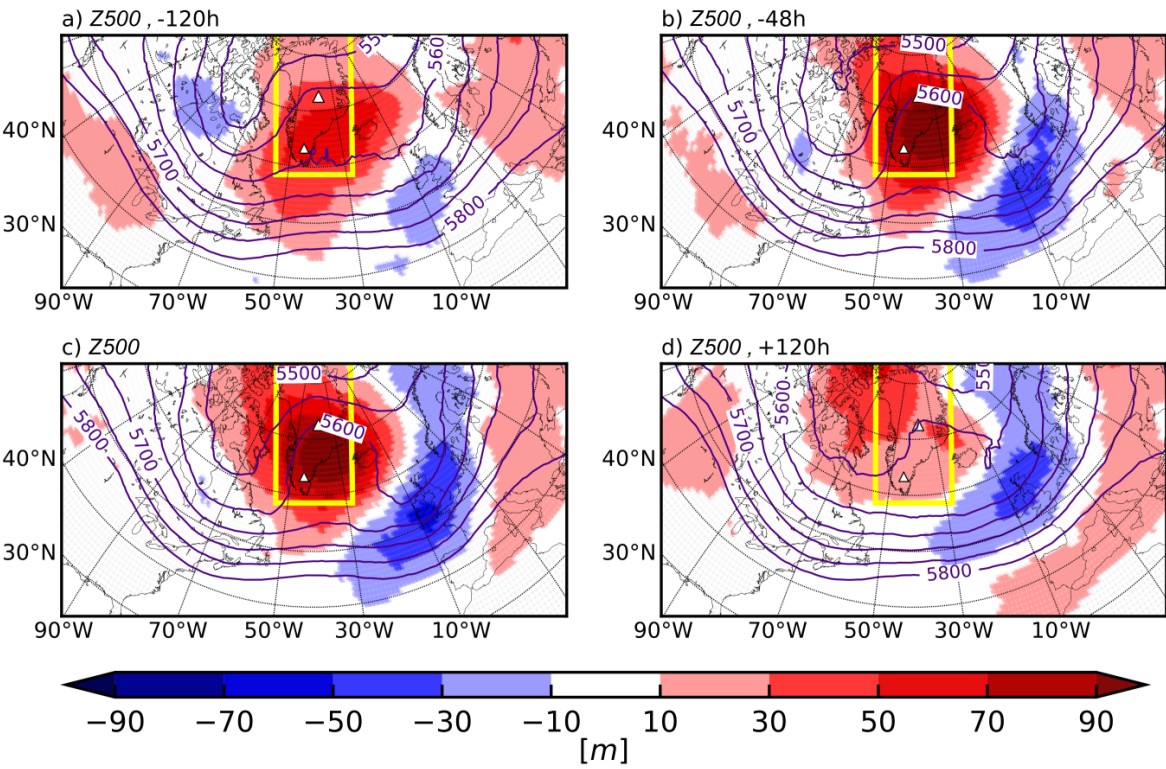

**Figure 8.** Median composites of 500 hPa geopotential height (*Z500*) in contours and their anomalies wrt. climatology in colors for different lags relative to all melt event time steps: (a) $lag = -120\,\mathrm{h}$, (b) $lag = -48\,\mathrm{h}$, (c) $lag = 0\,\mathrm{h}$, and (d) $lag = +120\,\mathrm{h}$. The yellow box indicates the location of the GrIS.

potential of 2.7 cm ice d$^{-1}$ (resulting from the heat of fusion of water, 333.55 J g$^{-1}$, and assuming ice at 0°C with a density of 917 kg m$^{-3}$).

## 4   Climatological analysis for melt events in 1979–2017

In this section we generalize the results from the EV69 case study by considering all 77 Greenland melt events. We present the synoptic situation during Greenland melt events (Sect. 4.1), the air masses associated with these events and their temperature modifications (Sect. 4.2), and finally precipitation, moisture, and radiation patterns over the GrIS and its subregions (Sect. 4.3).

### 4.1   Synoptic situation

We illustrate the synoptic situation related to all Greenland melt events in 1979–2017 by compositing *Z500* and its anomaly field, *Z500'*, relative to climatology (Fig. 8). The composites are calculated five and two days prior to the melt time steps, at



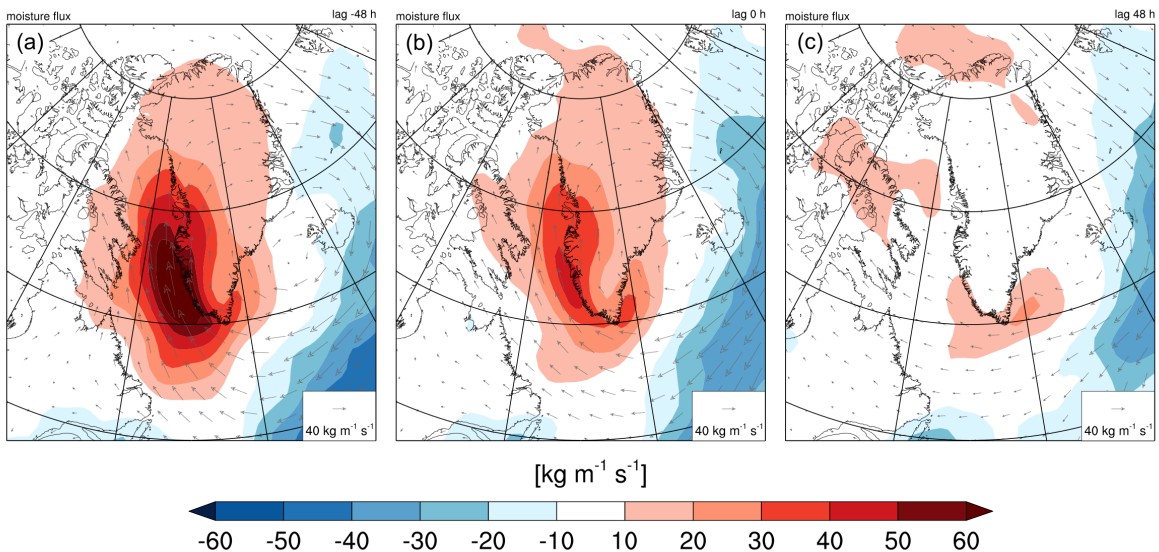

**Figure 9.** Median composites of total column horizontal water vapour transport anomalies ($TCVHT'$) wrt. climatology for different lags relative to melt event time steps: (a) $lag = -48\,\mathrm{h}$, (b) $lag = 0\,\mathrm{h}$, and (c) $lag = +48\,\mathrm{h}$.

the melt time steps, and five days later. It is important to note that some of the time steps entering lagged composites were themselves part of the respective melt event if the melt criterion is satisfied also at the lagged time. As for EV69, melt events are characterized by a dipole pattern of $Z500'$ with a positive anomaly centered over southeastern Greenland and a pronounced negative anomaly over northern Europe. A positive $Z500'$ of +50–70 m is typically present already 120 h before melt occurs, accompanied by troughs both upstream and downstream. This initial pattern is, thus, akin to the geopotential height anomalies

associated with cyclonic Rossby wave breaking and omega-type blocking (Fig. 8a; Liu and Barnes, 2015). Towards $lag = 0\,\mathrm{h}$, the anomalies transition into the dipole pattern with a strong ridge or cut-off anticyclone over Greenland and a pronounced trough over the British Isles and Scandinavia (Figs. 8b, c). $Z500'$ near Greenland peak at $>$+90 m around 24 h before melt occurs. Five days after a melt event, the positive $Z500'$ has shifted towards northwestern Greenland and the high Arctic (Fig. 8d). The dipole pattern is characteristic of the Greenland blocking regime (e.g., Grams et al., 2017), which projects negatively onto

the NAO index and positively onto the GBI.

This synoptic configuration provides favourable conditions for the poleward advection of warm and moist air masses from lower latitudes towards Greenland (Liu and Barnes, 2015), as reflected in the enhanced total column horizontal water vapour transport ($TCVHT$) especially along the southwestern GrIS (Fig. 9). The anomalous $TCVHT$ starts over the Labrador Sea

several days before the melt event (not shown) and the anomaly increases up to $>$+60 kg m$^{-1}$ s$^{-1}$ at $lag = -48\,\mathrm{h}$ while at the same time reaching the northern and central GrIS (Fig. 9a). The anomaly then gradually fades and vanishes completely in line with the weakening $Z500'$ after melt events (Figs. 9b, c). The centering of the positive geopotential height anomaly between



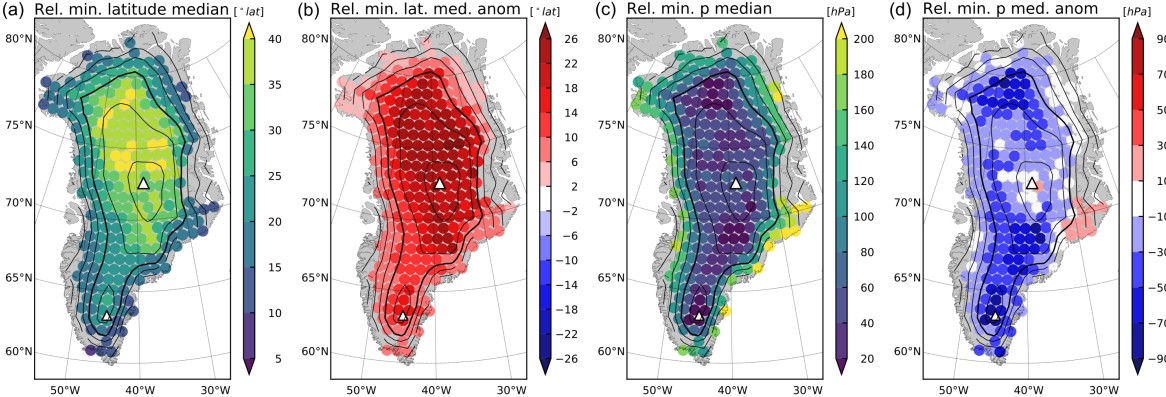

**Figure 10.** As Fig. 3 but for melt trajectories during all Greenland melt events.

southeastern Greenland and Iceland instead of over Greenland is a particularly important ingredient for the transport of moist-warm air parcels towards western Greenland and in an arc-shaped anticyclonic flow pattern across northwestern and northern

Greenland, as discussed already for EV69.

## 4.2   Lagrangian forward projection

The analyses presented in the following are analogous to those for EV69 (Sect. 3.2), but now for all Greenland melt events in JJA 1979–2017. Note, however, that melt at the most elevated parts of the GrIS is very rare, which is why the results near Summit strongly resemble those in EV69.

**4.2.1   Air mass origin**

Except for some arriving high on the GrIS, at $t = -192\,$h melt air masses are not anomalously warm compared to the local climatology $\theta_{cl}$ (Fig. S1a). They are, however, located much further south than usual, i.e., in a climatologically warmer region - more so for air masses arriving in region C and N and near Southdome (Fig. 10b). Coastal air masses previously move further poleward by 10° latitude and inland air masses by up to 26° latitude than normal summertime backward trajectories from these

location (Fig. 10a). At the same time, the melt trajectories reach less high levels prior to arriving on the GrIS and, therefore, experience less subsidence than climatological air masses (Fig. 10d). Comparing spatial patterns in Fig. 10, we find that, as during EV69, air masses ending in regions C, E, N, and S (subfigure in Fig. 2) share very similar transport characteristics (Sect. 3.2). N and S air masses reach the maximum elevation of their entire 8-day trajectory once they reach the GrIS (Fig. 10c). The only melt air masses influenced by anomalously strong descent arrive in region E after - embedded in the anticyclonic flow

- crossing the GrIS in an arc-like fashion (Fig. 10d).



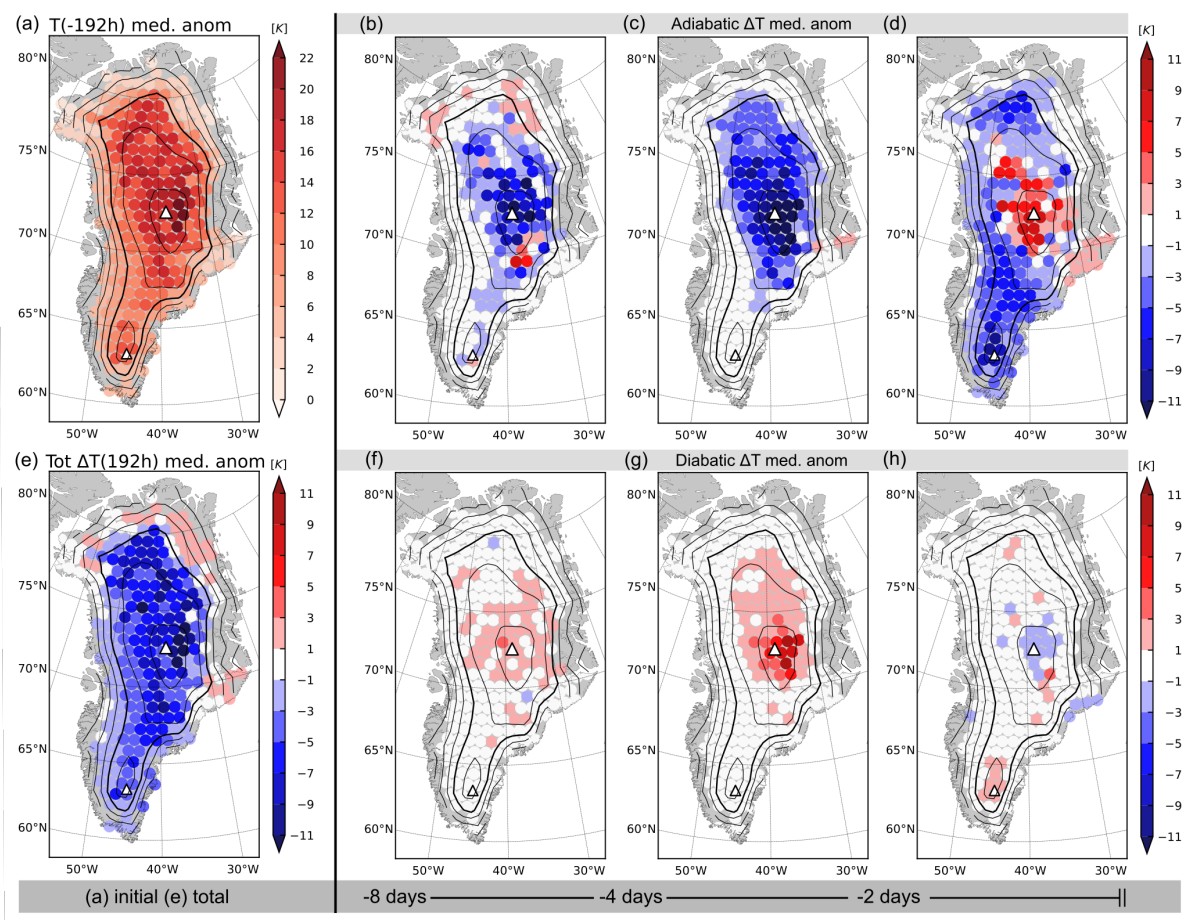

**Figure 11.** As Fig. 5 but for melt trajectories during all Greenland melt events.

### 4.2.2 Air mass evolution

Due to a climatologically warmer origin (lower elevation and/or lower latitude), all melt air masses are at $t = -192\,\mathrm{h}$ warmer wrt. the climatological summertime air mass arriving at the same location (Fig. 11a). There are no substantial local temperature anomalies at $t = -192\,\mathrm{h}$ (Fig. S1a). Thus, the positive anomalies of 10–20 K in Fig. 11a can be attributed to the unusual origin of air masses. The time series of adiabatic (Figs. 11b–d) and diabatic temperature change anomalies (Figs. 11f–h) along the trajectories during all melt events look very much alike the ones for EV69 (Sect. 3.2). C air masses ascend and cool adiabatically by a larger amount than the climatological air masses between $t = -144\,\mathrm{h}$ to $-48\,\mathrm{h}$ (Figs. 11b, c), while at the same time experiencing more diabatic heating (Figs. 11f, g). Furthermore, C and E air masses both show a stronger final descent reflected in enhanced adiabatic warming in the last two days (Fig. 11d). Finally, air masses ending in regions N and S ascend more wrt. the climatological summertime air masses within the last two days due to their advection towards sloping orography, that is





south- to northwesterly advection towards region N or southerly advection towards region S (Fig. 11d).

Overall, the anomalies are slightly weaker than for EV69 and show a stronger elevation dependency. Air masses at lower elevation have a thermodynamic history close to that of climatological air masses with slightly enhanced cooling for S, and en-

hanced warming for E and N air masses, respectively (Fig. 11e). In contrast, air masses ending in region C and near Southdome are more strongly cooled adiabatically, which is in part compensated by enhanced diabatic heating during the above-normal ascent. In total, however, most melt air masses experience stronger anomalous cooling during the eight-day period prior to arrival over the GrIS. As a consequence, only about 50-90 % of the higher initial temperature as seen in Fig. 11a remains when the air masses arrive on the GrIS.


In line with the characteristic regions shown in Fig. 2 and the spatial patterns identified in Figs. 10 and 11, we select several trajectory starting points between 2000–2500 m altitude representative of air masses ending in regions E, S, and N (see Fig. 12e). Furthermore, for region C, we consider melt trajectories arriving above 2500 m altitude. In the following we will consider the evolution of $T$, $\theta$, and $\theta_{cl}$ for trajectories arriving in each region. The air masses' evolution in $\theta - T$ space (Figs. 12a, c)

gives insight into the contributions of adiabatic temperature changes associated with vertical motion (changes along horizontal axis) and diabatic processes (changes along vertical axis). Furthermore, the evolution in $\theta - \theta_{cl}$ space (Figs. 12b, d) indicates when the final potential temperature anomalies emerge and it reveals the relative importance of transport from climatologically warmer regions towards Greenland (changes along horizontal axis) and diabatic processes (changes along vertical axis). For a more detailed discussion of this type of diagrams we refer to Papritz (2020).


The commonalities of all climatological air masses arriving over the GrIS include diabatic (radiative) cooling, as well as weak subsidence on the order of 25-50 hPa with associated adiabatic warming during the 8–2 d prior to arrival on the GrIS. This leads to a nearly isothermal temperature evolution in this period (Fig. 12a). Subsidence and poleward motion are both associated with transport from potentially warmer to a colder regions. Since this transport occurs at a rate exceeding that of

diabatic cooling, the trajectories acquire a weak potential temperature anomaly of +1–2 K until two days before arrival on the GrIS (Fig. 12b). Within the last one to two days, air masses ascend nearly adiabatically to the GrIS and those arriving in region E additionally descend during the final 12 h (Fig. 12a). During this final period, the potential temperature anomaly fades away and the climatological air mass arrives on the diagonal in the $\theta - \theta_{cl}$ space (Fig. 12b).

Now considering melt air masses, we see that they are initially around 5 K (E) to 18 K (C) warmer and evolve in a distinct way from the climatological summertime air masses (compare Figs. 12a, c). The differences are most striking for C air masses. During the first four days, these air masses remain at around 800 hPa (Fig. 12c) with no substantial local potential temperature anomaly (Fig. 12d). Then, within two to three days, they experience diabatic heating of around 6 K while ascending by nearly 250 hPa (Fig. 12c) and a potential temperature anomaly of more than 7 K forms (Fig. 12d). Note that the formation of the

potential temperature anomaly is about equally due to diabatic heating and transport into a climatologically colder region at the



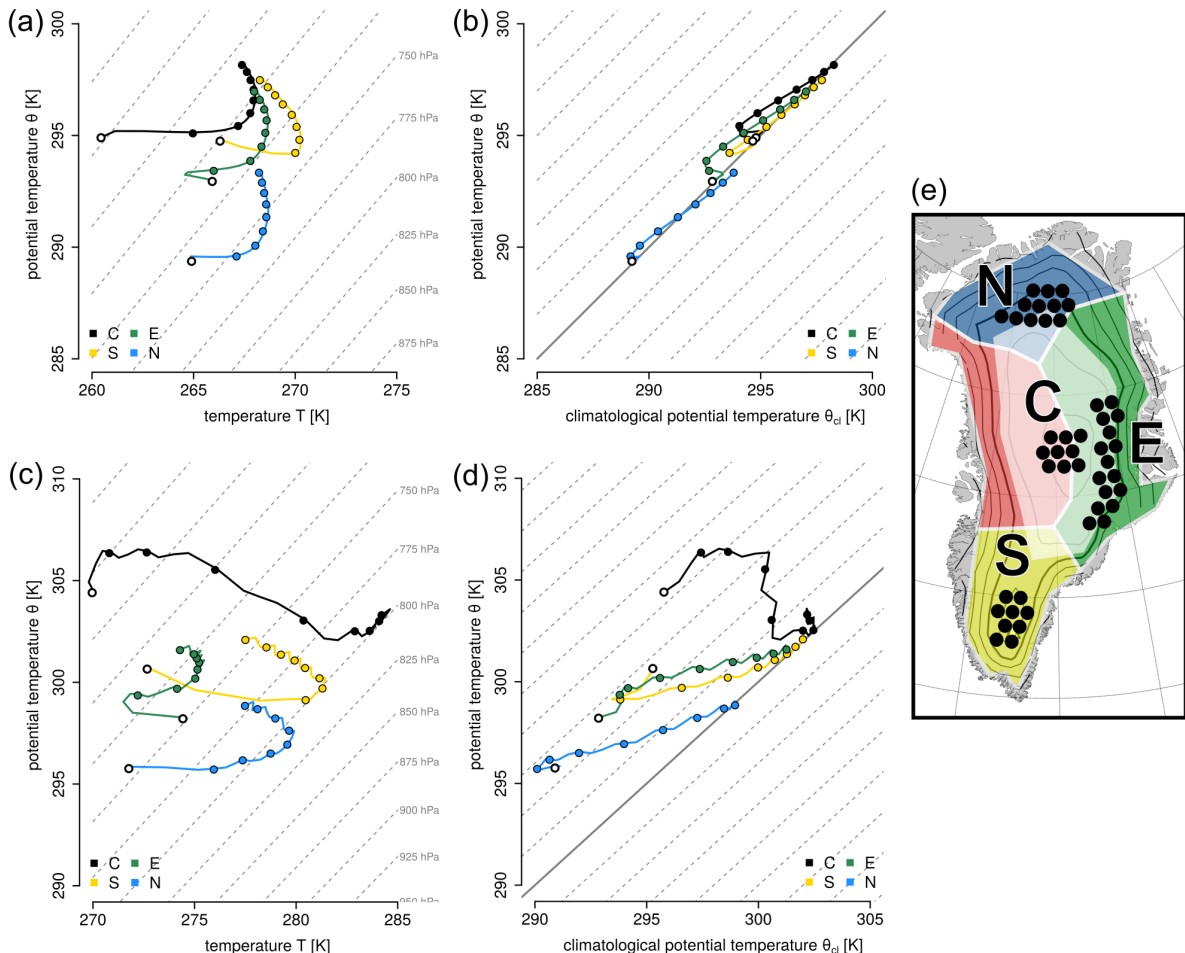

**Figure 12.** Thermodynamic evolution of backward trajectories from selected starting points (see panel e) representative of regions C, E, S, and N. Shown are medians for each trajectory category, summarized as one air parcel, in (a, c) $\theta - T$ and (b, d) $\theta - \theta_{cl}$ space. Grey dashed lines indicate isobars in intervals of 25 hPa (a, c) and isolines of constant potential temperature anomaly in intervals of 2 K (b, d). Panels (a, b) represent all climatological summertime air masses and (c, d) represent melt air masses only. Circles along the lines indicate time in 24-hour intervals, starting at $t = -192$ h and with an empty circle at the trajectory end point at $t = 0$ h. Note that the axis limits of plots (a–d) are not identical.

beginning and end of the backward trajectories. Overall, C melt air masses experience a strong cooling of 15 K in eight days, while their potential temperature slightly increases.

In contrast, E, N, and S air masses have an evolution in $\theta - T$ space that is more similar to climatological air masses (Fig. 12c).
Notable differences include reduced diabatic cooling and larger ascent and an associated decrease of temperature during the final two to three days, which is consistent with an origin at lower altitudes. Furthermore, the descent of air masses arriving





in region E is more pronounced. The decisive difference between melt event air masses and their climatological reference is, therefore, the much higher $T$ and $\theta$ values at $t = -192\,\mathrm{h}$. E, N, and S air masses show a similar evolution in $\theta - \theta_{cl}$ space as climatological air parcels, but with a stronger effect of transport that results in final potential temperature anomalies of about +3–5 K (Fig. 12d). The gradual increase of the potential temperature anomalies along with the similar temporal evolution as climatological air masses highlights the importance of the anomalous origin and the enhanced poleward transport for the anomalous nature of melt event air masses.

In summary, we note that air masses associated with melt over the GrIS have no initial potential temperature anomaly but they originate from climatologically warmer regions further to the South. The atmospheric circulation, characterized by a large positive geopotential anomaly over Southeast Greenland, then induces strong poleward transport towards the western GrIS, ascent, and latent heat release. Combined with the warmer origin of air masses, these diabatic temperature modifications are altogether responsible for the anomalously warm nature of the air masses when arriving over the GrIS. The importance of diabatic temperature modifications depends strongly on the altitude of the trajectory arrival position and are most important for air masses arriving near Summit.

### 4.3 Linkage to clouds and radiative effects

As in Section 3.3 for EV69, we analyze here the distribution of rainfall, total cloud liquid water, and surface radiation (Fig. 13), which are key quantities modulating the surface energy balance during the 77 observed melt events. As for EV69, there is a clear distinction between the regions west of the ice divide (including large parts of regions C, N, and S), which are exposed to the moist-warm anticyclonic inflow, and the eastern GrIS, which is located on the lee side of the of the ice divide and thus is affected to a much smaller degree by the increased total column horizontal water vapour transport (Fig. 9).

On average, more rain falls in regions NW, S, and W (Figs. 13a, e), especially around Southdome, where a strong shift of precipitation from snow to rain occurs (not shown). Everywhere except region E, there is less incoming solar radiation during melt events compared to climatology (Fig. 13g). In fact, in the higher elevated regions this roots in a shift towards higher *TCLW*, i.e., a phase shift to the liquid regime and potentially also more cloud cover (Fig. 13f). Even though *TCLW* is reduced at lower elevations along region W, there is sufficient cloud cover to reduce the incoming shortwave radiation. In contrast, in region E, related to descending air masses, little rain and reduced *TCLW* are a sign of clear-sky conditions, enabling *SSRD* to be very close to its maximum for this latitude and time of the year (Figs. 13a, f, g). Furthermore, the median total column water vapour anomaly is positive over the entire GrIS with an area-weighted average of $+2.8\,\mathrm{kg\,m^{-2}}$ (Fig. S3f). Despite the reduction of shortwave radiation west of the divide of the GrIS, the sum of net surface short- and longwave radiation is increased everywhere on the GrIS with an area-weighted average of $+7.3\,\mathrm{W\,m^{-2}}$ ($+2.1\,\mathrm{cm\,day^{-1}}$ melting potential calculated as in Sect. 3.3; Fig. 13h).



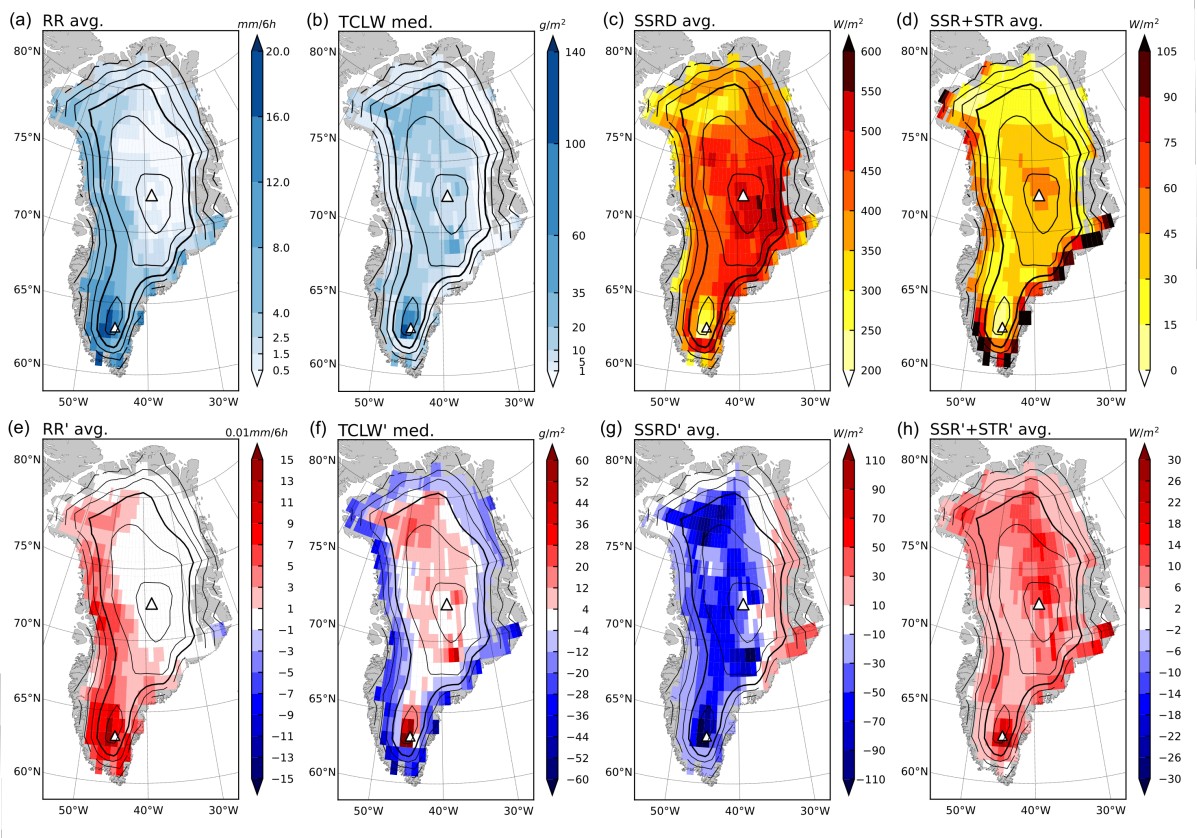

**Figure 13.** As Fig. 7 but for melt trajectories during all Greenland melt events.

In light of the disputed sign of the cloud radiative effect over the GrIS, we find that the warm-moist conditions during melt events as such increase the sum of incident short- and longwave radiation, even though contributions vary regionally as described above. The downward longwave radiation is on average increased by $+35.6 \, \text{W m}^{-2}$ and even more at high elevations, where the cloud water phase shifts, peaking at Southdome with an anomaly of $+99.6 \, \text{W m}^{-2}$ (Fig. S3h). The results here confirm the importance of moisture transport by the prevailing anticyclonic weather regime for Greenland melt events (Fig. 9) via the longwave radiative effects of water vapour and clouds as previously highlighted by Mattingly et al. (2016), Van Tricht et al. (2016), Ding et al. (2017) and Hofer et al. (2019). We conclude that during melt events:

1. moistening of the atmospheric column increases the incoming longwave radiation over the entire GrIS;

2. liquid clouds that were most prominent regions S, NW, and W go along with less incident shortwave radiation and thus reduce melt especially in the ablation area by preventing the strong shortwave ice albedo feedback (Box et al., 2012; Wang et al., 2019), but are highly dependent on the interplay of albedo and optical cloud properties (Van den Broeke et al., 2008);




3. rain, mostly affecting the same regions as (2), enhances melt (i) directly by its positive temperature, (ii) indirectly through the albedo feedback by darkening the surface (Fausto et al., 2016), and (iii) by thereby precondition the ice surface for following melt;

4. clouds in the high albedo accumulation region C have a net warming effect at the surface, especially with the observed higher liquid water content and if their optical thickness lies in the optimal intermediate range as during EV69 (Bennartz et al., 2013; Solomon et al., 2017).

Thus, our results indicate that during melt events, the synoptic circulation and transport patterns favour a distribution of cloud (and moisture) radiative effects that potentially enhance melt all over the GrIS because processes (1), (3), and (4) overcompen-
sate the competing shortwave cooling effect (2).

## 5   Discussion and Conclusions

We found 77 Greenland melt events during JJA 1979–2017 of more than one day duration (Question Q1, Sect. 1) by identifying melt of the Greenland Ice Sheet (GrIS) with a skin temperature $\geq -1°C$ from ERA-Interim data together with an elevation- and extent-based selection criterion (Sect. 2.2). These events became 60% more frequent and on average about two days longer
between the reference periods "recent past" (1986–2005) and "present day" (2005–2015) of the IPCC Special Report on the Ocean and Cryosphere in a Changing Climate (SROCC; Mintenbeck et al., 2020). Melt events longer than ten days, unprecedented in the "recent past", accounted for 18% of the "present day" melt events. Obviously, these trends follow from global warming (Johannessen et al., 2004), characterized by a pronounced warming in the Arctic known as Arctic amplification (e.g., Serreze and Barry, 2011). However, it is interesting to briefly discuss the importance of climate warming as compared to
circulation-induced warming for the occurrence and spatial extent of melt events.

The JJA near-surface potential temperature ($\theta_{10m}$) correlates well ($r = 0.66$) with the cumulative melt extent (CME) obtained from accumulating melt extent over all melt event time steps (see Section 2.2) in a given summer (Fig. 14a). This is especially noteworthy considering the asymmetry of the CME timeseries, which only varies in the presence of melt events but
not when they are absent (i.e., CME $\geq 0$). Despite the higher correlation of CME with climate warming ($r = 0.57$; Fig. 14b), there is a clear relationship between the seasonal circulation-induced $\theta_{10m}$ anomaly and CME ($r = 0.41$; Fig. 14c). Circulation can amplify warming by a factor of two, e.g., in summers 2010 and 2012, which belonged to the series of summers with persistent NAO-/GBI+ summer circulation anomalies (Fettweis et al., 2013; Hanna et al., 2018). Likewise, circulation can also offset climate warming such as in summers 2009 and 2015. The exceptional melt event EV69, discussed here as a case study,
was part of - and contributing to - the warmest summer on record (2012; $\theta'_{10m} = +2.6 \pm 0.6$ K). EV69 is a textbook example of a Greenland melt event as most of the general dynamical and thermodynamic characteristics of melt events were strongly pronounced.

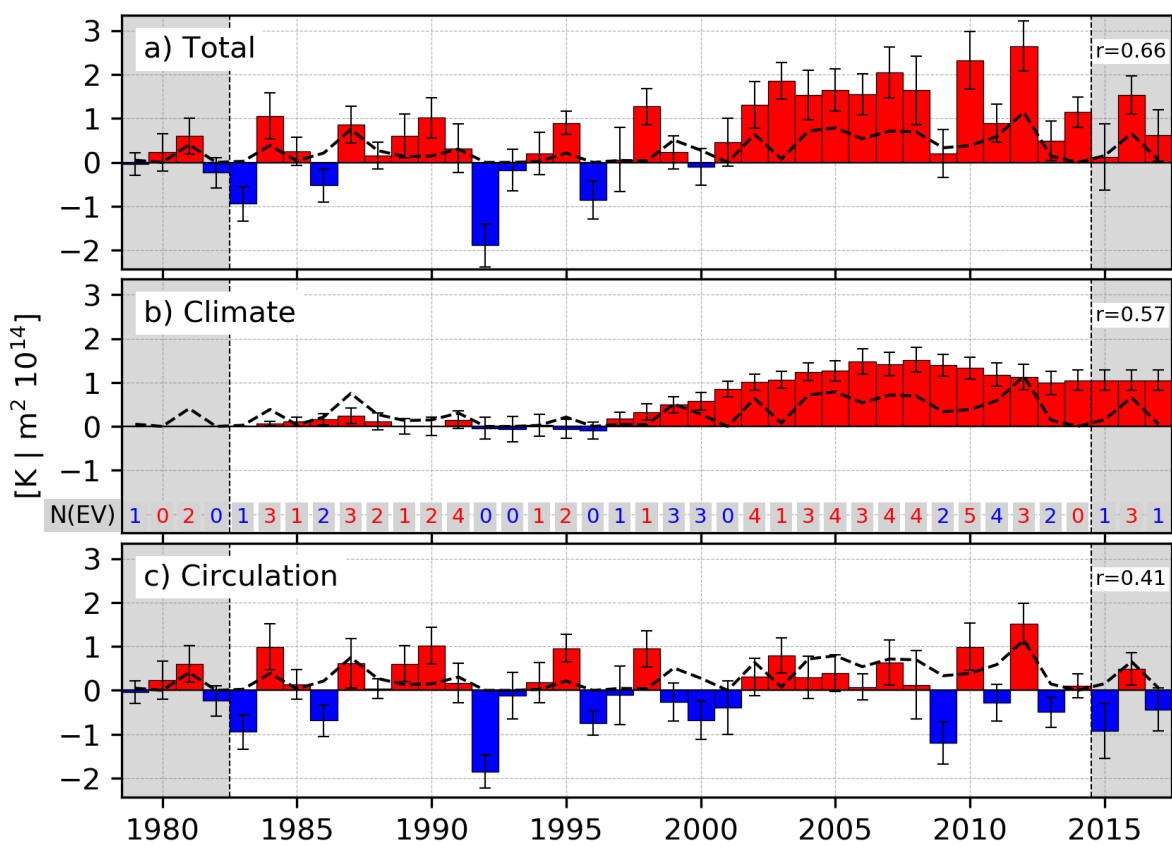

**Figure 14.** A time series of the (a) total, (b) climatological (9-yr centered running mean), and (c) circulation-induced (i.e., total minus climatological) near-surface potential temperature anomaly, $\theta'_{10m}$, temporally averaged over JJA and spatially (standard deviation shown by whiskers) over the GrIS. The time series are shown wrt. the climatology at the beginning of the time series (1979/1983), and their correlation with the cumulative melt extent during melt events (dashed line) is given in the upper right of each panel. Numbers in (b) indicate the number of Greenland melt events in this summer, colored according to (c). Due to the running mean used in (b), the decomposition of the temperature anomaly shown in (b) and (c) is only meaningful in the period 1983–2014.

The most prominent synoptic characteristic of Greenland melt events is a upper-level ridge or tropospheric cut-off with its center located southeast of Greenland (Q2, Sect. 1). Despite the anomalously strong final descent of air masses arriving in central and eastern Greenland during these events, large-scale subsidence and adiabatic warming within the anticyclonic flow anomaly is of very little importance for Greenland melt events. This is opposed to lower tropospheric warm extremes in the central Arctic (Binder et al.; Papritz, 2020) and in mid-latitude heat waves (Bieli et al., 2015; Zschenderlein et al., 2019). The location of the geopotential height anomaly southeast of Greenland is favourable for inducing a southerly flow and enhanced total column horizontal water vapour transport to the West and towards the southern tip of Greenland. As these air masses impinge on Greenland's orography, they are forced to ascend, accompanied by cloud formation and precipitation, subsequently





followed by anticyclonic transport across the GrIS and eventually descent along the eastern slope of the ice sheet.

The two most important processes contributing to the warm anomaly of air masses of Greenland melt events are (Q3, Sect. 1):

1. **Transport:** Melt event air masses originate from a region that is 15 K warmer than climatological air masses. At their origin eight days prior to arrival in Greenland, melt event air masses are, however, not generally anomalously warm. Hence, it is their origin at lower latitude and/or lower altitude and the subsequent rapid meridional transport of up to $40°$ latitude that are decisive for their final temperature anomaly. During transport to the GrIS, the melt air masses cool more than usual, i.e., by $\sim$15 K, which keeps the trajectories arriving closest to the surface just above the critical threshold to
induce melt. As we found, the warm anomaly associated with air masses that arrived near Summit during EV69 arose due to strong meridional transport to Greenland and did not result from a pre-existing warm anomaly such as associated with a heat wave in the Great Plains of North America (Hoerling et al., 2014; Neff et al., 2014; Bonne et al., 2015).

2. **Latent heat release:** As the GrIS has an average elevation of more than 2000 m, airstreams ascend either dynamically or orographically along their 8-day trajectory. This ascent occurs at the poleward edge of a band of prominent horizontal
moisture transport. Latent heat release during ascent and, consequently, cloud formation is contributing substantially to the final warm anomaly of air masses arriving over the plateau region. Both processes, meridional transport and latent heating, are most pronounced for the high-elevation, central regions of the GrIS. Air masses causing melt higher on the GrIS come from warmer and more southerly regions and experience diabatic warming and adiabatic cooling that deviates more from the climatological summertime air parcel (Q4, Sect. 1). Melt air masses of the northern and southern GrIS
undergo ascent later along their 8-day trajectory and have an origin at lower levels than air masses of the central GrIS.

We further find that the melt events go along with increased total column water vapour (everywhere), related to the enhanced poleward moisture transport, and a phase change of cloud water and precipitation from ice to liquid forming where airstreams ascend over the southern tip of Greenland, along the west coast, as well as in the North of the GrIS (Q3, Sect. 1). The resulting net radiative anomaly contributes positively to surface melt during Greenland melt events (GrIS-wide area-weighted average
melting potential of 2.1 cm ice day$^{-1}$), especially in high-elevated regions and the Northwest of the GrIS. Clear-sky conditions prevail over the eastern GrIS, resulting in reduced total column water and enhanced incoming shortwave radiation. In contrast, the enhanced liquid water content in the elevated regions in the South, West and North of the GrIS lead to a reduction of incident shortwave radiation. Yet, the net radiation anomaly is positive due to enhanced longwave radiation. Generally, downward longwave radiation is a key element in triggering surface melt in Greenland and the remaining Arctic (Mortin et al., 2016; Lee
et al., 2017), and enhanced poleward moisture transport improves the simulation of Arctic clouds and near-surface temperature (Baek et al., 2020). The dynamical and thermodynamic characteristics of melt event air masses found here, confirm the importance of moisture transport as a result of the long-range transport of air masses from the South towards Greenland for (i) inducing latent heating along the trajectory, and (ii) causing a positive cloud radiative effect over the GrIS.



As the large-scale dynamics is found to be the key driver of Greenland melt events, the understanding of upper-tropospheric ridges and blocks and their development and lifespan is highly relevant to Greenland's climate, GrIS mass loss (Hanna et al., 2014; Van den Broeke et al., 2017), and global sea-level rise (Van den Broeke et al., 2016; Box et al., 2018). The dynamical understanding of blocks (Pfahl et al., 2015; Steinfeld and Pfahl, 2019) and heat extreme-related upper-tropospheric ridges (Zschenderlein et al., 2020) now includes the important role of upstream latent heating for establishing and maintaining the

negative potential vorticity anomalies in the upper troposphere. The representation of those processes in climate models is yet uncertain. More generally, global climate models are yet not able to capture the strong and persistent NAO- circulation anomalies of recent years (Fettweis et al., 2012, 2013). If these changes are the result of natural variability, long-term trends predicted by the models could still be trustworthy, as the model performance may mainly be limited by the internal variability of the climate system (Fischer et al., 2013; Knutti and Sedláček, 2013). In the long run, Greenland blocking is not predicted to

change significantly towards the end of this century (e.g., Gillett and Fyfe, 2013). If, however, the current decrease in summer NAO is a manifestation of systematic circulation changes associated with global warming, the ability of today's climate models to simulate future trends in the North Atlantic circulation is questionable, and GrIS mass loss at the end of this century could be underestimated by a factor of two (Delhasse et al., 2018). Given the importance of upper-tropospheric ridges and blocks, and associated transport of moist-warm air for Greenland melt events, future work should, therefore, focus on their representation,

life-cycle and trends in climate models.

*Code and data availability.* All results are based on ERA-Interim data, which can be downloaded from ECMWF (https://apps.ecmwf.int/ datasets/data/interim-full-daily/levtype=sfc/), and analyzed with two additional tools: LAGRANTO (Wernli and Davies, 1997; Sprenger and Wernli, 2015) and clim-ei (Sprenger et al., 2017). Scripts used for the analyses and plotting, mostly written in Python 3.7, are available on request from the authors.

*Author contributions.* MH performed most of the analyses, supported by LP, and wrote a first version of the manuscript based on his MSc thesis (Hermann, 2019). All authors contributed to the design of the study, the interpretation of the results and the writing.

*Competing interests.* The authors declare that they have no conflict of interest.

*Acknowledgements.* We thank the ECMWF for providing access to ERA-Interim data, and Michael Sprenger (ETH Zurich) for technical support with the LAGRANTO and clim-ei tool. This study has been partially funded by the H2020 European Research Council (INTEXseas;

grant no. 787652).





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
