# Peer review of "A Lagrangian Analysis of the Dynamical and Thermodynamic Drivers of Greenland Melt Events during 1979–2017"

_Weather and Climate Dynamics, 2020_

## Referee Comment (RC1) · Xavier Fettweis (Referee) · 2 Jun 2020

This paper discusses the origin of air masses generating melt events over the ice sheet by focusing first on July 2012 as example and after to the 1979-2017 climatological mean. While this paper is quite complex (high scientific level) and not easy to understand after a quick first reading, it is very original, with a clear aim and certainly deserves to be accepted for publication. I have however several (minor) remarks:

- When the author are discussing the origin of air masses, it is not clear which vertical level is considered? A height of 20, 40, 60 hPa above the Greenland ice sheet surface is mentioned. Which one is used? How are the authors sure that the considered level

is not in the boundary layer and then, impacted by the katabatic winds for example?

- When the impact on the net surface solar radiation is discussed (eg: Figs 7 and 13), the presented results are depend of the ERA-Int resolution (100km) which is not enough to represent the ablation zone (with a lot of lower surface albedo than snow and a width typically lower than 100km). Moreover, I'm not sure that ERA-Int is able to represent the bare ice albedo (0.3-0.5) when the ablation zone is larger than 100km. Therefore, this issue should be absolutely discussed in the manuscript and the conclusions discussed in Section 4.3 (lines 452-465) are in fact only valid in the accumulation zone as the ablation zone is not really represented here by ERA-Interim. In the ablation zone, as discussed in Hofer et al. (2017), the shortwave anomalies drive the melt and clouds have a cooling effect.

- While Summer 2019 is not studied here, I would like to mention that the 01-AUG-2019 big melt event was generated by air masses coming from Europe and having crossed North-Atlantic (Tedesco and Fettweis, 2020). Such an origin in a melt event is not mentioned here suggesting that such origin is very exceptional and such a event deserves to be studied in further studies.

- The results presented here are based on ERA-Interim. I don't ask to redo this study using the new generation ERA5 reanalysis, but the use of ERA5, improving a bit the representation of the near surface condition over the ice sheet (Delhasse et al., 2020) and available at a higher resolution (30km) more suitable to represent the ablation zone, could be also mentioned in the perspective.

Reference:

- Tedesco, M. and Fettweis, X.: Unprecedented atmospheric conditions (1948–2019) drive the 2019 exceptional melting season over the Greenland ice sheet, The Cryosphere, 14, 1209–1223, https://doi.org/10.5194/tc-14-1209-2020, 2020.

- Delhasse, A., Kittel, C., Amory, C., Hofer, S., van As, D., S. Fausto, R., and Fettweis, X.: Brief communication: Evaluation of the near-surface climate in ERA5 over the Greenland Ice Sheet, The Cryosphere, 14, 957–965, https://doi.org/10.5194/tc-14-957-2020, 2020.

---

## Referee Comment (RC2) · Stefan Hofer (Referee) · 9 Jun 2020

The comment was uploaded in the form of a supplement:
https://wcd.copernicus.org/preprints/wcd-2020-16/wcd-2020-16-RC2-supplement.pdf
* * *

---

## Author Comment (AC1) · 17 Jul 2020

The comment was uploaded in the form of a supplement:
https://wcd.copernicus.org/preprints/wcd-2020-16/wcd-2020-16-AC1-supplement.pdf

———————————————————

---

## Author Response (AR2)

**A Lagrangian Analysis of the Dynamical and Thermodynamic Drivers of Greenland Melt Events during 1979–2017**

*Reply to both reviewers by Mauro Hermann, Lukas Papritz, and Heini Wernli*

We highly appreciate the fruitful and detailed comments of both reviewers, which greatly support investigating the link of atmospheric dynamics and Greenland Ice Sheet (GrIS) surface melt. The greatest concern of both reviewers was the ability of ERA-Interim (1° horizontal resolution) to represent the narrow GrIS ablation zone (20-100 km wide), where the majority of GrIS surface melt occurs. The reviewers highlighted that with "Greenland melt events", as defined in our methods section, we refer to a special category of melt events, i.e., to large-scale melt events affecting the high elevation accumulation zone of the ice sheet. Also, the conclusions drawn from our results on (cloud) radiative effects are only valid for large-scale melt events over the accumulation zone and are not representative of the processes causing melt in the ablation zone. We fully share this view and are grateful for this important critique. The revised version now carefully explains the distinction between ablation zone melt processes and the high-elevation, large-scale melt events, and emphasizes that our study only considers the second type of melt events.

In the following, we group reviewer comments regarding this main concern, and answer them and the remaining comments separately. This document is further supplemented with a latexdiff-pdf (in the same format as the first version of the manuscript) that highlights the modifications made by point-by-point tracking, and to which we will refer in the answers below whenever a specific line number is indicated. Later in the resubmission procedure, we will also submit this revised version of the manuscript.
* * *
**Reviewer comments 1 by Xavier Fettweis (XF) and 2 by Stefan Hofer (SH) on main concern:**

**SH:** *Scientific assessment:* While the general presentation of the study is to be applauded, there are some distinct methodological limitations that render this study likely to be only valid for a subset of the Greenland Ice Sheet due to the inability to resolve the ablation zone with the ERA-Interim data (i.e. the accumulation zone), and only for a specific type of widespread, long-lasting (24h+) melt events, that also affect the bright interior of the Greenland Ice Sheet, where melt dynamics are significantly different to the ablation zone.
However, as it currently stands the authors draw conclusions on the contribution of clouds, humidity and airmass advection to the general Greenland Ice Sheet melt, which simply cannot be resolved spatially (~100 km resolution). Due to the temporal and elevation focus on widespread melt events, which in themselves can only be driven by large-scale anomalous advection of warm and humid airmasses, it seems that the generalization of the results is likely limited. Overall, more than 80% of all ice melt occurs in the ablation zone - where weather station data shows absorbed shortwave radiation to be the main driver of melt - which is unfortunately not resolved by ERA Interim data in this study.
Because the data used doesn't resolve the darker ablation zone, and the initial selection of melt events likely greatly skews the analysis towards longwave and humidity driven melt events at high elevations, the presented results in their current form cannot be presented as being generally valid to all Greenland melt and likely needs a more nuanced presentation throughout the manuscript.
However, I hope that my comments will encourage the authors to slightly rethink some conclusions of their manuscript, after which this paper will be a welcomed contribution to the growing set of novel Greenland Ice Sheet climate literature.

**Authors:** Thank you, this comment is extremely helpful, as are the specific comments following below. We fully agree that the findings of our study do not apply to the typical melt in the ablation zone. Instead our study focuses on exceptional, large-scale melt events that affect, in particular, also the elevated accumulation zone, such as during the melt event in summer 2012. In the revised manuscript, we better explain the melt event selection, and the limitations of ERA-Interim to resolve the ablation zone and, therefore, the dominant melt forcing as follows:

1. We refer to the events defined according to our methodology as "large-scale melt events" or "melt events" instead of "Greenland melt events", referring to the terms introduced in the methods (Sect. 2.2). In the title, we replace "Greenland melt events" with "large-scale Greenland melt events". This accounts for the nature and definition of the melt events, as we expect a "large-scale" melt event to fulfill our extent- and elevation-based criteria. We will not highlight the duration of 24h+ in this terminology due to the procedure of connecting melt time steps, which are separated by up to 24h.
2. When introducing the trajectory setup (Sect. 2.3., L148), we highlight that the vast majority of the starting points lies in the accumulation zone, shown by the percentage of starting points above 1000/1500/2000m (90%/76%/56%).
3. We introduce our research questions more clearly (especially in L91-93), such that it becomes clear that we focus on – and thus, the answers to the research questions are valid for – large-scale melt events including the accumulation zone.
4. ERA-Interim resolution: Our intention is to resolve the large-scale atmospheric flow prior to the defined melt events, and to understand how this then affects the GrIS surface conditions in different regions (S, W, N, E, C instead of ablation/accumulation zone). We fully agree that the ablation area is not at all resolved in ERA-Interim, and our conclusions should only concern either the trajectory history prior to melt events or accumulation zone radiative effects. However, these melt events mostly cover large parts of the ablation area as well, which is why it would not be entirely correct to only refer to the accumulation zone. Of course, when specifically looking into cloud and moisture radiative effects, our analysis cannot attribute a cause of melt to ice in the ablation area (see below). But the ablation area is affected by the synoptic situation described in our study as well. Thus, we tried to better balance the fine line of framing our conclusions correctly. They should clearly not attribute a cause to ablation area melt, but they should neither exclude the effect of the atmospheric circulation on the entire GrIS during large-scale melt events.

**SH:** *Title*: Maybe include "of large-scale" or "extreme" melt events, because for the reader it seems that the authors focus primarily on a very specific subset of melt events, that might not represent the general physical mode of "normal" melt.

**Authors:** Thank you for this suggestion, which we like very much. We changed the title to "A Lagrangian Analysis of the Dynamical and Thermodynamic Drivers of *Large-Scale* Greenland Melt Events during 1979–2017". While the melt events selected in our study are certainly exceptional, we prefer, however, not to include "extreme" in the title given that we find 77 melt events in less than 40 years.

**SH:** *Introduction*: The introduction was very enjoyable to read and is a very concise account of the current state of the relevant Greenland literature. Around L90 – the main questions the authors want to answer. Q1) "How often did melt events occur over the GrIS during 1979-2017?" Maybe mention that the authors focus on widespread, long-lasting melt events that affect the bright ablation zone. Q3) Also potentially highlight that the radiative effects and air mass modifications are valid for your chosen subset of melt events, not the "normal" GrIS melt. Q4) "Does the answer to Q2 and Q3 differ for subregions of the GrIS?" → For the reader the most interesting question would likely be does it differ for the accumulation zone melt events vs. the ablation zone melt events where more than 80% of all melt occurs and where the physical drivers are significantly different due to the difference in albedo.

**Authors:** We agree with your suggestions. We reword our main research questions (L100-104) to:

- Q1) How often did *large-scale* melt events occur over the GrIS during 1979–2017?
- Q2) What are the synoptic flow configuration and the air mass pathways during *these* melt events?
- Q3) Which thermodynamic air mass modifications and radiative effects *over the GrIS accumulation zone* caused these melt events?
- Q4) Does the answer to Q2 and Q3 differ for subregions of the GrIS *accumulation zone*?

We now discuss the difference of our melt event results (longwave radiative forcing) and the typical shortwave-dominated ablation zone melt in the discussion section (L579 onwards) based on the results of known literature (Hofer et al., 2017, Wang et al., 2019, Izeboud et al., 2020). In addition, we address ablation area melt separately in the introduction (L50-65).

**SH:** L94 ff – Era Interim data: At 1° resolution the ERA Interim data does give the authors one pixel in the SW of the GrIS where the ablation zone is at its widest and even less where the ablation zone lies in steeper terrain. Both is not sufficient to resolve the ablation zone. For the reader it seems that the study design is robust to answer Q2 mentioned in the introduction, but Q1, Q3 and Q4 can only be answered for the accumulation zone.

**Authors:** Absolutely correct, see previous comments.

**SH:** L111 For the reader it likely needs to be made clearer throughout the manuscript that the authors are dealing with a specific subset of widespread melt events, and not the ablation zone melt dynamics that contribute most to GrIS melt.

**Authors:** We now highlight our focus on *large-scale* melt events in the title and abstract, instead of using the too generic term "Greenland melt events". The introduction more clearly differentiates between radiative processes affecting the ablation vs. the accumulation zone and better emphasizes the focus of our study (see above). Furthermore, we emphasize our melt event definition in the method section (L118, L120, L129, L148) and remind the reader in the result subsections (L311, L342-345, L371, L455).

**SH:** L319 "In this section we generalize the results from the EV69 case study by considering all 77 Greenland melt events." I think this statement is somewhat misleading. The authors are not generalizing to "all" melt events, but still only to a subset of widespread and longwave driven melt events that reach up to high elevations and last longer than 24h. For the reader this nuance is lacking through most parts of the following and preceding discussion of results and should be added throughout, also that ERA I does not resolve the ablation zone.

**Authors:** Already in the previous form of the manuscript, we chose a specific terminology "Greenland melt events" referring to events as defined in the method section, and carefully made use of this terminology throughout the manuscript. We acknowledge the nuance mentioned here, and will carefully define the term "melt events" in the methods section to avoid confusion with statements about general "Greenland melt events". Also, we change the terminology in the specific sentence to "all 77 large-scale melt events" (L342). We prefer this terminology over somewhat lengthy terms such as "large-scale melt events affecting the GrIS accumulation zone" which would make the manuscript quite cumbersome to read.

Regarding the comment on ERA-Interim resolution, we completely agree (see answer to the first remaining comment by XF below).

**SH:** L374 "… but now for all Greenland melt events in JJA 1979-2017" See previous comment on why this might be an overstatement.

**Authors:** Agree, see above. Changed to "for all 77 large-scale melt events in JJA 1979-2017" in L371.

**XF:** When the impact on the net surface solar radiation is discussed (eg: Figs 7 and 13), the presented results are depend of the ERA-Int resolution (100km) which is not enough to represent the ablation zone (with a lot of lower surface albedo than snow and a width typically lower than 100km). Moreover, I'm not sure that ERA-Int is able to represent the bare ice albedo (0.3-0.5) when the ablation zone is larger than 100km. Therefore, this issue should be absolutely discussed in the manuscript and the conclusions discussed in Section 4.3 (lines 452-465) are in fact only valid in the accumulation zone as the ablation zone is not really represented here by ERA-Interim. In the ablation zone, as discussed in Hofer et al. (2017), the shortwave anomalies drive the melt and clouds have a cooling effect.

**SH:** L426 and following section "Linkage to clouds and radiative effects" The authors likely need to take into account the limitations of their approach here. The used data simply doesn't allow to answer the question of what is driving most of the GrIS melt, given that a great majority of melt occurs in the ablation zone that isn't resolved here. Additionally, the chosen subset of melt events skews the conclusion drawn for the contribution of radiation and clouds to quite obviously longwave driven melt events in the bright accumulation zone of the Greenland Ice Sheet. For the reader this section needs quite a bit more of a nuanced assessment, especially from L448 onwards, where the authors conclude that the longwave radiative effect of clouds and humidity is the main contributor to melt and enhanced furthers by anticyclonic circulation.

**Authors:** Both comments highlight the need for a short discussion of the limitations of our approach. We now outline in the introduction (L77, L91-93) and method section (see above) that our focus lies on (i) the large-scale air mass transport and modifications before large-scale melt events, and (ii) on exceptional, large-scale melt events affecting the high-albedo/elevation accumulation zone.

Then, most importantly, the conclusions drawn in Sect. 4.3 were strongly reduced, such that it becomes clear to the reader that our main focus lies on the subsections before (air mass modifications during large-scale melt events). The main result drawn from Sect. 4.3, namely that these air masses enhance longwave radiative forcing as known for the GrIS accumulation zone, is highlighted more accurately (L474-499).

Furthermore, we will discuss the limitations of ERA-Interim (horizontal resolution, boundary layer representation) in the discussion and conclusion section (see first remaining comment by XF).

**SH:** L523 ff "longwave radiation is a key element in triggering surface melt in Greenland and the remaining Arctic". The results here are only representative for the accumulation zone melt dynamics and not Greenland in general. Unfortunately, this statement cannot be concluded from the presented analysis.

**Authors:** Yes → changed to "in the GrIS accumulation zone and the remaining Arctic" in L572.

**SH:** L528 ad "ii) causing a positive cloud radiative effect" – Yes, but this can only be answered here for the interior of the GrIS where the surface albedo is high and therefore the cloud radiative dynamics are significantly different to the ablation zone (additional to a skewed subset of melt events). Just today a paper has been published that shows that the CRE is negative over the GrIS ablation zone during summer, and positive over the accumulation zone. (https://agupubs.onlinelibrary.wiley.com/doi/epdf/10.1029/2020GL087315).

**Authors:** Thank you, this study is very useful for discussing the cloud radiative effects and especially for differentiating between ablation and accumulation zone melt. We include it to put our results of accumulation zone melt into perspective (L499, L580, L584).

**XF:** The results presented here are based on ERA-Interim. I don't ask to redo this study using the new generation ERA5 reanalysis, but the use of ERA5, improving a bit the representation of the near surface condition over the ice sheet (Delhasse et al., 2020) and available at a higher resolution (30km) more suitable to represent the ablation zone, could be also mentioned in the perspective.

**Authors:** We now provide an outlook in the discussion and conclusion section that addresses this issue (L579-589), including a reference to Delhasse et al. (2020). There, we mention the limitations of ERA-Interim, and refer to ERA5 and/or regional climate models (MAR, RACMO), which are better suited for studying local melt processes, for comparison to in situ measurements, and for surface energy fluxes thanks to their higher resolution of the steep topography and improved albedo.

**Remaining comments by XF (RC1):**

This paper discusses the origin of air masses generating melt events over the ice sheet by focusing first on July 2012 as example and after to the 1979-2017 climatological mean. While this paper is quite complex (high scientific level) and not easy to understand after a quick first reading, it is very original, with a clear aim and certainly deserves to be accepted for publication. I have however several (minor) remarks:

- When the authors are discussing the origin of air masses, it is not clear which vertical level is considered? A height of 20, 40, 60 hPa above the Greenland ice sheet surface is mentioned. Which one is used? How are the authors sure that the considered level is not in the boundary layer and then, impacted by the katabatic winds for example?

  **Authors:** We average these three layers in our analyses (LFP maps, e.g., Fig. 3) to account for potential deficiencies of ERA-Interim in representing the boundary layer over the GrIS. In our study we are interested in the origin, transport pathways, and modifications of air masses present near the surface during large-scale melt events (highlighted now in L152). Choosing 20, 40, and 60 hPa above the surface as starting positions allows us to select air masses in the proximity of the surface, while at the same time avoiding potential deficiencies of ERA-Interim in representing the detailed structure of the boundary layer above the GrIS [a highly important layer for surface melt, e.g., Ohmura et al. (2001)]. Furthermore, we assume the representation of the near-surface flow in ERA-Interim to be sufficient for our study design, as we focus on air mass modifications on a time-scale of ten days and air mass movement of several thousand kilometers.

  Still, we fully agree that it is important to include a discussion of the resolution and limitations of ERA-Interim in the discussion and conclusion section (L579-589). Our intention is not to resolve the detailed processes in the ablation zone, nor to make use of the (insufficient) abilities of ERA-Interim to represent boundary layer processes such as katabatic drainage flows. We therefore clearly point out that we focus on the large-scale atmospheric transport/modification of air masses arriving close to the GrIS surface during large-scale melt events, i.e., over the accumulation zone (see above). Our set of melt events could be the starting point for studies of the detailed boundary layer processes during these events using RCM or in situ data.

- While Summer 2019 is not studied here, I would like to mention that the 01-AUG-2019 big melt event was generated by air masses coming from Europe and having crossed North-Atlantic (Tedesco and Fettweis, 2020). Such an origin in a melt event is not mentioned here suggesting that such origin is very exceptional and such an event deserves to be studied in further studies.

**Authors:** This is an interesting point. We supplement our manuscript with a figure showing the trajectory density of all 77 large-scale melt events together, at different times before arrival over melting parts of the GrIS during a melt event (see below, based on the gridding tool v2.4.2; Škerlak, 2014). There are two tongues of increased trajectory density stretching over Scandinavia and Great Britain. From the cyclonic movement of the tongue between t=−192h and t=−96h (Figs. S2a-c), it seems likely that some of the melt events show a similar air mass origin and transport pathway as the 1 August 2019 event, likely linked to a cyclone near Great Britain and an anticyclone over (West) Greenland. Nevertheless, the bulk of the air masses clearly originates from west of Greenland, indicating that transport pathways as during the 1 August 2019 event are rather exceptional, which we focus on in L379-383.

[Figure]

**Remaining comments by SH (RC2):**

In this study, Hermann et al. present a Lagrangian analysis of one specific melt event in summer 2012 and 77 long-lasting melt events between 1979-2017. The authors use a threshold value for the surface temperature of -1°C to identify melt from the ERA Interim dataset. The authors focus on more extreme, widespread melt events with more than 5% of Greenland melting simultaneously, melt occurring above 2000m and only focus on melt events lasting for more than 24 hours. Additionally, the authors also use ERA Interim data to try and establish the underlying drivers of melt events (surface radiative fluxes, liquid water content and others). Furthermore, the authors establish a novel way to identify the main areas of synoptic-scale airmass advection prior and during these 77 more pronounced melt events.

Overall, this study is very well written and consists of a set of nicely presented figures. The approach the authors use is scientifically novel and has the potential to shed some light on the question of how circulation and advection patterns influence Greenland melt.

**Specific comments by SH:**

L135 Is there a previous study that looks at the capabilities of ERA-Interim to accurately model the GrIS boundary layer, given that all the parcels of the authors are starting in the lowermost 500m of the atmosphere?

**Authors:** We are not aware of a study that systematically evaluated the boundary layer over the GrIS in general. However, several studies investigated the representation of specific parameters such as near surface winds, surface temperature, or long- and shortwave radiative fluxes in the Arctic in general and over the GrIS in particular against in situ observations. These studies agree that among state-of-the-art reanalyses, ERA-Interim performs particularly well (Chen et al. 2011, Cox et al. 2014, Lindsay et al. 2014, Wesslén et al. 2014, Fettweis et al. 2017). Furthermore, Delhasse et al. (2020) found certain improvements of ERA5 compared to ERA-Interim over the GrIS in terms of surface variables such as 2m temperature or 10m winds, but noted that the differences are not significant.

For the purpose of our study, it is especially important that the large-scale wind fields are properly represented such as to capture the air mass transport onto the GrIS appropriately. Moore et al. (2013) and Oltmanns et al. (2014) compared wind fields with station data and found generally a good agreement over Greenland, emphasizing in particular also the capability of ERA-Interim to represent downslope windstorms. However, Moore et al. (2013) also noted that – common to coarse resolution models – turbulence parameterizations are too diffusive, leading to too deep boundary layers and too weak temperature inversions.

By starting trajectories not just from the surface but instead from 20, 40, and 60 hPa agl, we capture the bulk of near-surface air masses arriving over the GrIS. This is appropriate for our study focusing on the large-scale spatial and temporal evolution (10 days, thousands of kilometers) of the air masses arriving over the GrIS.

L195 How did the authors identify the "dry intrusion" near Newfoundland? Isentropic potential vorticity based analysis?

**Authors:** We did not define this airstream as dry intrusion → Changed "descended in the dry intrusion" to "descended cyclonically in the low near Newfoundland" in L212.

L270 ff Discussion about previously warm and humid airmasses losing their warm anomaly but staying humid. Might it potentially not be better to look at a variable that combines temperature and humidity, such as equivalent potential temperature, to define an airmass? Is there a specific reason why the authors chose not to?

[Figure]

**Authors:** This paragraph should shed light on whether the potential temperature anomaly over the U.S. Great Plains was the start of the potential temperature anomaly over Greenland. Equivalent potential temperature would indeed be useful as a tracer for humid air masses, but it would obscure warming by condensation of water vapor, which appears important for the formation of the final temperature anomaly. We use the LAGRANTO trajectory tool to trace the air mass, and then look into why it became anomalously warm (it is important to note that THanom indicates the potential temperature anomaly of the air mass wrt. the local climatology.

The figure added here additionally illustrates the potential temperature and pressure evolution along the heat wave trajectories. We can conclude that the strong increase in THanom by almost 10K between t=−120h and t=−48h can be attributed to: (i) Net 6K diabatic heating (panel c) due to latent heat release during ascent of ~150hPa and overcompensated radiative cooling (panel b), (ii) transport into a climatologically 4K colder region (of course Greenland is more than 4K colder than the U.S. East coast, but the air masses also ascend to higher altitudes). The amount of diabatic heating fits to the 6K diabatic heating of C air masses apparent from the TH-T diagram (Fig. 12c).

L284 "cloud formation" Until this point the authors did not look at clouds specifically, so some of the changes in radiative fluxes could be due to phase change in existing clouds alone and not just due to extra cloud formation.

**Authors:** Good point, changed "cloud formation and latent heating" → "latent heating from condensation of water vapor" (L303).

L289 How robust is the phase partitioning in the atmospheric column, i.e. is the total column liquid water in ERA-I reliable in high-latitudes? Maybe there is a study to cite that looks at this specifically.

**Authors:** Overall, ERA-Interim represents the liquid water path in the Arctic relatively well and at times better than models with a more complex microphysics scheme (cf. Wesslén et al. 2014) – now mentioned in L313. However, there are certain known biases especially over the GrIS. For instance, Bennartz et al. (2013) found that in summer ERA-Interim underestimates the occurrence of thin liquid clouds at Summit.

L295 Interestingly, it seems that in the NW of the GrIS there is even a negative anomaly of liquid water content in lower elevation areas, and only when the airmass ascents further it suddenly develops a positive liquid water content anomaly. Any ideas of why that is, given that the airmass in itself likely has a higher specific humidity content than in the climatology overall?

**Authors:** This is indeed an interesting phenomenon, as the NW region showed melt to high elevations during most of EV69. However, lower elevations were approached by normal to slightly colder air masses in the beginning of EV69 (see Fig. 1b). There are two air streams at play: N1 representing the air stream arriving at higher elevations (warm-moist) and N2 being somewhat representative of air masses reaching the lower elevations from the Northwest (and also Northeast; Fig. 2). N1 trajectories reach saturation in the very end, and the further they ascend, the further they carry liquid cloud water, the positive TCLW anomaly, and the negative SSRD anomaly inland. Conditions over the lower NW region are mostly sunnier than climatology, i.e., clouds are thinner or still contain ice at higher altitudes (the SSRD' median is positive in this region (Fig. 7g shows the average)). The transition of negative to positive TCLW' thus represents the interaction/boarder of the two air streams.

L295-308 Maybe the authors could mention that the discussion here is still focusing at one specific melt event? Sometimes this wasn't clear for the reader.

**Authors:** We will add in the introducing sentence of this paragraph "during EV69" in L317.

occurs, optically thick clouds have a cooling effect and their reduction in the past 20 years coincided with enhanced surface mass loss (Hofer et al., 2017). Contrarily in the high albedo accumulation zone, clouds were found to have a warming effect due to downward longwave radiation, which was particularly pronounced in summer 2012  Furthermore, moist-warm conditions and  liquid clouds are not only instantaneous drivers of melt, but their effect also accumulates over time to precondition surface melt, on daily (Solomon et al., 2017), seasonal (Park et al., 2015) and annual time scales (Tedesco et al., 2013).

The transport of anomalously warm and humid air masses is a key driver of  large-scale melt events over the GrIS.  Warm-moist air implies strong  non-radiative energy fluxes into the ice, such as for example in July 2012 (Hanna et al., 2014; Fausto et al., 2016). During that period, the transport of warm air from a concurrent heat wave over North America (Hoerling et al., 2014) to the GrIS was suggested to be directly related to two melt events peaking at 98% and 79% melt extent (Neff et al., 2014). Additionally, the involved moisture transport from the western subtropical North Atlantic triggered cloud radiative effects favorable for  surface melt in the accumulation zone (Neff et al., 2014; Bonne et al., 2015). Optically thin liquid clouds enhanced downward longwave radiation, still letting shortwave radiation penetrate, and enabled surface melt over the normally dry  inland plateau (Bennartz et al., 2013). Additionally, air temperature near the surface directly affects the downwelling longwave radiative fluxes since the bulk of these is emitted in the lowermost kilometer of the atmosphere (Ohmura, 2001).  As such large-scale melt events are expected to become more frequent, we will focus on  air masses arriving during such periods near the GrIS surface.

Three processes can, in principle, contribute towards the formation of a warm  anomaly of airstreams reaching the GrIS (Papritz, 2020), namely the transport of an already warm air mass from a climatologically warmer region towards the GrIS, adiabatic compression during subsidence, and heating by diabatic processes. The latter comprises radiation, latent heat release in clouds, and turbulent surface fluxes (e.g., Holton and Hakim, 2012). In particular, subsidence is known to be an essential contributor to mid-latitude heat waves (Bieli et al., 2015; Zschenderlein et al., 2019) and warm anomalies in the high Arctic (Ding et al., 2017; Wernli and Papritz, 2018; Papritz, 2020). Furthermore, turbulent surface fluxes over the ocean are typically limited in summer due to the small surface-atmosphere temperature gradient.

 Considering the relevance of the atmospheric circulation for the variability of Greenland's near-surface climate, the goal of our study is to improve our understanding of the atmospheric dynamical processes leading to  melt episodes that cover large parts of the GrIS accumulation zone. This knowledge is  important given the strong impact of such exceptional melt events on the surface mass balance and the expected increase of and the ice sheet's melt extent, mass loss, and contribution to global sea

level rise. Furthermore, it might shed light on climate models struggling to simulate the observed circulation anomalies (e.g., Fettweis et al., 2012, 2013). More precisely, this study has two main objectives: First, we want to go beyond case studies and investigate large-scale melt events systematically in the period of 1979–2017. Still, the well-studied and most  extensive melt event of July 2012 will serve as an excellent example to illustrate our methods and findings. Second, we aim to investigate the history and thermodynamic evolution of air masses associated with  large-scale melt events with the aid of Lagrangian backward trajectories. This approach will enable us to answer the following questions:

Q1) How often did large-scale melt events occur over the GrIS during 1979–2017?

Q2) What are the synoptic flow configuration and the air mass pathways during these melt events?

Q3) Which thermodynamic air mass modifications and radiative effects over the GrIS accumulation zone caused these melt events?

Q4) Does the answer to Q2 and Q3 differ for subregions of the GrIS accumulation zone?

**2 Data and methods**

**2.1 ERA-Interim data**

This study is based on ERA-Interim reanalysis data from the European Centre for Medium-Range Weather Forecasts (ECMWF; Dee et al., 2011). The data is available every 6 h from 1979 to 2017, on 60 vertical levels and interpolated to a grid with a horizontal grid spacing of $1°$. The reanalysis data serves as best estimate of the past atmospheric state on the synoptic scale, which is why we implicitly refer to it as the actual state of the atmosphere. As climatologies of the variables used for Eulerian analyses (Table 1), we compute 10 d-averages of the 6-hourly data centered on the respective calendar day over the entire period 1979–2017. Note that for radiation, we use fields with the same time of day only to account for the daily cycle. We use the ice outline after Zwally et al. (2012) to separate ice from land grid cells in Greenland. Only grid cells with a center inside the ice outline are classified as ice grid cells, which leads to  a GrIS area of 1.73 million $km^2$, which is slightly larger (+0.7%) than observed (Zwally et al., 2012).

**2.2 Melt event definition**

As previous studies focused on single  large-scale melt events, such as in July 2012 (e.g., Nghiem et al., 2012; Bennartz et al., 2013; Tedesco et al., 2013; Neff et al., 2014; Bonne et al., 2015), there is yet no generally accepted definition of  such melt events for climatological studies. We define them as follows: The occurrence of surface melt is approximated by a skin temperature ($SKT$) greater or equal to $-1°C$, as in earlier studies (e.g., Nghiem et al., 2012). A time step is interpreted as part of a melt event if at least 5% of the total GrIS surface area is melting and  located above 2000 m elevation ("melt time step"), to distinguish melt events from the typical summer melt in the GrIS ablation area. In order

to avoid splitting of a - from a dynamical point of view - coherent melt event due to the pronounced diurnal temperature cycle, we include non-melt time steps when identifying coherent melt events. This is done as follows: intermediate non-melt and melt time steps are connected in time to yield melt events with the starting (end) date defined as the first (last) time step when melt was detected, but not preceded (followed) by melt for more than 24 hours. The thresholds of 5% and 2000 m were chosen with hindsight, such that a reasonable maximum melt event duration of around two weeks and a sufficiently large sample size of 77 melt events results. Events shorter than 24 hours are neglected. To summarize, we focus on melt events that cover a wide area of the GrIS accumulation zone and hereafter refer to them as "large-scale melt events" or "melt events" for simplicity.

The 77  melt events in 1979–2017 (Tables S1–S3) lasted between 1.25 d and 16.25 d and on average $4.1 \pm 3.4$ d (Table 2). Surface melt during short events typically covered around a third of the GrIS at maximum. On average, about half $(44.6 \pm 10.7\%)$ of the GrIS was melting at the time of maximum extension of the event. The three melt events affecting the largest ice area were EV69 (94.8%), EV35 (83.9%) and EV70 (70.3%) in early July 2012, June 2002 and end of July 2012, respectively. EV69 is the most closely investigated melt event in the literature, where surface melt occurred up to the highest ERA-Interim grid point at 3175 m. Considering all events, the maximum elevation with surface melt was $2692 \pm 193$ m. The maximum two-meter temperature at the most elevated grid point experiencing melt averaged slightly below 0°C.

**2.3 Backward trajectories**

 We use the Lagrangian framework to investigate air mass modifications, the underlying physical processes and general flow structures. The Lagrangian analysis tool LAGRANTO (Wernli and Davies, 1997; Sprenger and Wernli, 2015) basically solves the trajectory equation (Eq. 1) numerically.

$$\frac{D\mathbf{x}}{Dt} = \mathbf{u}(\mathbf{x}) \tag{1}$$

where $\mathbf{x}$ is the position of an individual air parcel and $\mathbf{u}$ the 3D wind vector. We use 3D ERA-Interim wind fields to calculate kinematic backward trajectories from pre-defined starting locations and trace a set of variables along the trajectories (Table 3). In the domain defined by the 519 ice grid points (Sect. 2.2) trajectories are started equidistantly every 80 km in the horizontal, resulting in 267 starting points per height level. Most of the trajectory starting points are located over the GrIS accumulation zone, as 90%, 76%, and 56% of the starting points' ground level lies above 1000 m, 1500 m, and 2000 m, respectively. In the vertical, trajectories start at three near-surface layers in the lowermost ~500 m of the atmosphere (20, 40 and 60 hPa above ground level~~, i. e., from the lowermost ~500 m of the atmosphere, resulting in 3 × 267 starting points~~), representing air masses that exert a strong surface forcing. During the evaluation, we consider trajectories from all three layers to get a more robust estimate of the properties of typical air masses near the GrIS. 
[revised manuscript text omitted]

**3.3 Linkage to clouds and radiative effects**

The characteristic airstreams during EV69 (C, E, N1, N2, S; Fig. 2) likely played an important role in modulating the spatial distributions of rain and snowfall, cloud liquid water, and radiative fluxes over the GrIS, which in turn had a strong impact on the melt potential. To illustrate these inter-linkages, we consider rain rate (*RR*), total column liquid water (*TCLW*), surface shortwave downward radiation (*SSRD*) and the sum of net surface short- and longwave (thermal) radiation (*SSR+STR*; Figs. 7a–d), as well as their anomalies wrt. the 1979–2017 summer climatology (Figs. 7e–h) for melt time steps at the respective grid points. We note that our results are based on the specific selection of large-scale melt events (Sect. 2.2) and ERA-Interim, whose resolution ($\sim 100\,\mathrm{km}$) is not resolving the narrow ($20 - 100\,\mathrm{km}$) ablation zone topography and surface processes correctly, i.e., mainly address characteristics of the accumulation zone. Still, the investigated variables and effects are relatively well represented in the Arctic in ERA-Interim (Wesslén et al., 2014; Wang et al., 2019).

[revised manuscript text omitted]
 from ERA-Interim at 1° horizontal resolution, i.e., being most meaningful over the GrIS accumulation zone (Fig. 13). As for EV69, there is a clear distinction between the regions west of the ice divide (including large parts of regions C, N, and S), which are exposed to the moist-warm anticyclonic inflow, and the eastern GrIS, which is located on the lee side of the of the ice divide and thus is affected to a much smaller degree by the increased total column horizontal water  vapor transport (Fig. 9).

On average, more rain falls in regions NW, S, and W (Figs. 13a, e), especially around Southdome, where a strong shift of precipitation from snow to rain occurs (not shown). Everywhere except region E, there is less incoming solar radiation during melt events compared to climatology (Fig. 13g). In fact, in  regions above 2000 m this roots in a shift towards higher *TCLW*, i.e., a phase shift to the liquid regime and potentially also more cloud cover (Fig. 13f). Even though *TCLW* is reduced at  elevations below 2000 m along region W, there is sufficient cloud cover to reduce the incoming shortwave radiation. In contrast, in region E, related to descending air masses, little rain and reduced *TCLW* are a sign of clear-sky conditions, enabling *SSRD* to be very close to its maximum for this latitude and time of the year (Figs. 13a, f, g). Furthermore, the median total column water  vapor anomaly is positive over the entire GrIS with an area-weighted average of $+2.8 \, \text{kg m}^{-2}$ (Fig. S4f). Despite the reduction of shortwave radiation west of the divide of the GrIS, the sum of net surface short- and longwave radiation is increased everywhere on the GrIS with an area-weighted average of $+7.3 \, \text{W m}^{-2}$ ($+2.1 \, \text{cm day}^{-1}$ melting potential calculated as in Sect. 3.3; Fig. 13h).

To summarize, we find  an increase of net surface radiation during large-scale melt events, even though long- and shortwave contributions vary regionally. The ERA-Interim-derived results are, however, representative for the bright GrIS accumulation zone only. The warm-moist anticyclonic flow conditions relate to an increase of downward longwave radiation  by on average $+35.6 \, \text{W m}^{-2}$. In the upper accumulation zone, where the cloud water phase  is shifted wrt. climatology, this anomaly peaks at $+99.6 \, \text{W m}^{-2}$ near Southdome (Fig.

 S4h). Our findings underline the importance of  longwave radiative forcing - initially induced by meridional moisture transport - for melt events affecting large parts of the GrIS accumulation zone (Fig. 9), as previously highlighted by Mattingly et al. (2016), Van Tricht et al. (2016),

485

1.

2.

490 3.

4.

495

 Wang et al. (2019), Hofer et al. (2019), and Izeboud et al. (2020).

500 ## 5  Discussion and Conclusions

**5.1  Large-scale Greenland melt events**

We found 77 large-scale Greenland melt events during JJA 1979–2017 of more than one day duration (Question Q1, Sect. 1) by identifying melt of the Greenland Ice Sheet (GrIS) with a skin temperature $\geq -1°C$ from ERA-Interim data together with an elevation- and extent-based selection criterion (Sect. 2.2). These events became 60% more frequent and on average about
505 two days longer between the reference periods "recent past" (1986–2005) and "present day" (2005–2015) of the IPCC Special Report on the Ocean and Cryosphere in a Changing Climate (SROCC; Mintenbeck et al., 2020). Melt events longer than ten days, unprecedented in the "recent past", accounted for 18% of the "present day" melt events. Obviously, these trends follow from global warming (Johannessen et al., 2004), characterized by a pronounced warming in the Arctic (e.g., Serreze and Barry, 2011), and large-scale melt events are expected to cover
510 the entire ice sheet in the near future (Box et al., 2012). However, it is interesting to briefly discuss the importance of climate

warming as compared to circulation-induced warming for the occurrence and spatial extent of melt events.

The JJA near-surface potential temperature ($\theta_{10\mathrm{m}}$) correlates well ($r = 0.66$) with the cumulative melt extent (CME) obtained from accumulating melt extent over all melt event time steps (see Section 2.2) in a given summer (Fig. 14a). This is especially noteworthy considering the asymmetry of the CME timeseries, which only varies in the presence of melt events but not when they are absent (i.e., CME $\geq 0$). Despite the higher correlation of CME with climate warming ($r = 0.57$; Fig. 14b), there is a clear relationship between the seasonal circulation-induced $\theta_{10\mathrm{m}}$ anomaly and CME ($r = 0.41$; Fig. 14c). Circulation can amplify warming by a factor of two, e.g., in summers 2010 and 2012, which belonged to the series of summers with persistent NAO-/GBI+ summer circulation anomalies (Fettweis et al., 2013; Hanna et al., 2018). Likewise, circulation can also offset climate warming such as in summers 2009 and 2015. The exceptional melt event EV69, discussed here as a case study, was part of - and contributing to - the warmest summer on record (2012; $\theta'_{10\mathrm{m}} = +2.6 \pm 0.6\,\mathrm{K}$). EV69 is a textbook example of a  large-scale melt event as most of the general dynamical and thermodynamic characteristics of melt events were strongly pronounced.

**5.2 Large-scale air mass transport and transformations contributing to melt events**

The most prominent synoptic characteristic of  large-scale melt events is a upper-level ridge or tropospheric cut-off with its center located southeast of Greenland (Q2, Sect. 1). Despite the anomalously strong final descent of air masses arriving in central and eastern Greenland during these events, large-scale subsidence and adiabatic warming within the anticyclonic flow anomaly is of very little importance for  the identified melt events. This is opposed to lower tropospheric warm extremes in the central Arctic (Binder et al., 2017; Papritz, 2020) and in mid-latitude heat waves (Bieli et al., 2015; Zschenderlein et al., 2019). The location of the geopotential height anomaly southeast of Greenland is favourable for inducing a southerly flow and enhanced total column horizontal water  vapor transport to the West and towards the southern tip of Greenland. As these air masses impinge on Greenland's orography, they are forced to ascend, accompanied by cloud formation and precipitation, subsequently followed by anticyclonic transport across the GrIS and eventually descent along the eastern slope of the ice sheet.

The two most important processes contributing to the warm anomaly of air masses of  large-scale melt events are (Q3, Sect. 1):

1. **Transport:** Melt event air masses originate from a region that is $15\,\mathrm{K}$ warmer than climatological air masses.  However, at their origin eight days prior to arrival in Greenland, melt event air masses are  not generally anomalously warm. Hence, it is their origin at lower latitude and/or lower altitude and the subsequent rapid meridional transport of up to $40°$ latitude that are decisive for their final temperature anomaly. During transport to the GrIS accumulation zone, the melt air masses cool more than usual, i.e., by $\sim15\,\mathrm{K}$, which keeps  those arriving closest to the surface just above the critical threshold to induce melt. As we found, the warm anomaly associated with air

masses that arrived near Summit during EV69 arose due to strong meridional transport to Greenland and did not result from a pre-existing warm anomaly such as associated with a heat wave in the Great Plains of North America (Hoerling et al., 2014; Neff et al., 2014; Bonne et al., 2015).

2. **Latent heat release:** As the GrIS has an average elevation of more than 2000 m, airstreams ascend either dynamically or orographically along their 8-day trajectory. This ascent occurs at the poleward edge of a band of prominent horizontal moisture transport. Latent heat release during ascent and, consequently, cloud formation is contributing substantially to the final warm anomaly of air masses arriving over the accumulation zone. Both processes, meridional transport and latent heating, are most pronounced for the  upper accumulation zone, i.e., the central GrIS. The higher in the GrIS accumulation zone air masses arrive, the warmer and more southerly  is their region of origin, and the more the experienced diabatic warming and adiabatic cooling  deviates from the climatological summertime air parcel (Q4, Sect. 1). Melt air masses of the northern and southern GrIS undergo ascent later along their 8-day trajectory and have an origin at lower levels than air masses of the central GrIS.

**5.3 Air mass impact on the GrIS**

We specifically investigated the air mass history and the related near-surface conditions during large-scale melt events, which affect large parts of the GrIS accumulation zone (Sect. 2.2). During such events, total column water  vapor is increased, associated with an enhanced poleward moisture transport  and with a phase change of cloud water and precipitation from ice to liquid. The latter forms where airstreams ascend over the southern tip of Greenland, along the west coast, as well as in the North of the GrIS (Q3, Sect. 1).  Therefore, the incident shortwave radiation is reduced over the western and central GrIS, while the opposite applies for the clear-sky regions east of the ice divide. The net surface radiation anomaly is positively contributing to surface melt  over the entire GrIS accumulation zone, and is only in the very East dominated by shortwave radiation. In contrast, the enhanced liquid water content in the  South, West and North of the GrIS  accumulation zone causes an anomalously strong longwave radiative forcing, relating to the net cloud warming observed in these regions (Wang et al., 2019). This is a key  process for surface melt in  the GrIS accumulation zone and the remaining Arctic (Mortin et al., 2016; Lee et al., 2017), and enhanced poleward moisture transport improves the simulation of Arctic clouds and near-surface temperature (Baek et al., 2020). The dynamical and thermodynamic characteristics of melt event air masses found here, confirm the importance of poleward moisture transport as a result of the long-range transport of air masses from the South towards

Greenland for (i) inducing latent heating along the trajectory, and (ii) causing a positive cloud radiative effect over the GrIS accumulation zone.

580    Over the low albedo ablation zone, where the majority of GrIS surface mass loss occurs, the cloud radiative effect is typically cooling as shortwave radiation drives surface melt via the more efficient albedo-melt feedback (Hofer et al., 2017; Izeboud et al., 2020). While ERA-Interim is able to reproduce the warming cloud radiative effect over the accumulation zone during summer (Wang et al., 2019), the ablation zone is insufficiently represented in our data set and, therefore, precluded from the analysis. First and foremost, an accurate representation of the low ablation area albedo in summer $(0.3 - 0.5)$ would be crucial to

585    determine the surface melt resulting from the synoptic forcing during melt events (Izeboud et al., 2020). Also, the steep topography of the $20 - 100\,\mathrm{km}$ wide ablation zone is not resolved in ERA-Interim ($\sim 100\,\mathrm{km}$ grid spacing). There, the impact of the presented melt air masses could in future work be studied with regional climate models such as MAR (Fettweis et al., 2017), or RACMO (Noël et al., 2016), which are run on the kilometer-scale including more sophisticated surface schemes. Also, the latest generation of reanalysis data, ERA5, with $0.5°$ horizontal resolution improves the simulated near-surface climate over

590    Greenland to some degree (Delhasse et al., 2020).
           As the large-scale

**5.4   Importance of upper-tropospheric ridges**

As the atmospheric dynamics is found to be the key driver of Greenland large-scale melt events, the understanding of upper-tropospheric ridges and blocks and their development and lifespan is highly relevant to Greenland's climate, GrIS mass loss

595    (Hanna et al., 2014; Van den Broeke et al., 2017), and global sea-level rise (Van den Broeke et al., 2016; Box et al., 2018). The dynamical understanding of blocks (Pfahl et al., 2015; Steinfeld and Pfahl, 2019) and heat extreme-related upper-tropospheric ridges (Zschenderlein et al., 2020) now includes the important role of upstream latent heating for establishing and maintaining the negative potential vorticity anomalies in the upper troposphere. The representation of those processes in climate models is yet uncertain. More generally, global climate models are yet not able to capture the strong and persistent NAO- circulation

600    anomalies of recent years (Fettweis et al., 2012, 2013). If these changes are the result of natural variability, long-term trends predicted by the models could still be trustworthy, as the model performance may mainly be limited by the internal variability of the climate system (Fischer et al., 2013; Knutti and Sedláček, 2013). In the long run, Greenland blocking is not predicted to change significantly towards the end of this century (e.g., Gillett and Fyfe, 2013). If, however, the current decrease in summer NAO is a manifestation of systematic circulation changes associated with global warming, the ability of today's climate models

605    to simulate future trends in the North Atlantic circulation is questionable, and GrIS mass loss at the end of this century could be underestimated by a factor of two (Delhasse et al., 2018). Given the importance of upper-tropospheric ridges and blocks, and associated transport of moist-warm air for Greenland large-scale melt events, future work should, therefore, focus on their representation, life-cycle and trends in climate models.

*Code and data availability.* All results are based on ERA-Interim data, which can be downloaded from ECMWF (https://apps.ecmwf.int/ datasets/data/interim-full-daily/levtype=sfc/), and analyzed with two additional tools: LAGRANTO (Wernli and Davies, 1997; Sprenger and Wernli, 2015) and clim-ei (Sprenger et al., 2017). Scripts used for the analyses and plotting, mostly written in Python 3.7, are available on request from the authors.

*Author contributions.* MH performed most of the analyses, supported by LP, and wrote a first version of the manuscript based on his MSc thesis (Hermann, 2019). All authors contributed to the design of the study, the interpretation of the results, and the writing.

*Competing interests.* The authors declare that they have no conflict of interest.

*Acknowledgements.* We thank the reviewers Xavier Fettweis and Stefan Hofer for their constructive feedback on the first version of the manuscript. We acknowledge the ECMWF for providing access to ERA-Interim data, and Michael Sprenger (ETH Zurich) for technical support with the LAGRANTO and clim-ei tool. This study has been partially funded by the H2020 European Research Council (INTEXseas; grant no. 787652).